# Transcriptional regulation of endothelial cell behavior during sprouting angiogenesis

Hyun-Woo Jeong[1,2], Benjamín Hernández-Rodríguez [3], JungMo Kim[1,2], Kee-Pyo Kim[4], Rocio Enriquez-Gasca [3], Juyong Yoon[4], Susanne Adams[1,2], Hans R. Schöler [2,4], Juan M. Vaquerizas [3] & Ralf H. Adams [1,2]

Mediating the expansion of vascular beds in many physiological and pathological settings, angiogenesis requires dynamic changes in endothelial cell behavior. However, the molecular mechanisms governing endothelial cell activity during different phases of vascular growth, remodeling, maturation, and quiescence remain elusive. Here, we characterize dynamic gene expression changes during postnatal development and identify critical angiogenic factors in mouse retinal endothelial cells. Using actively translating transcriptome analysis and in silico computational analyses, we determine candidate regulators controlling endothelial cell behavior at different developmental stages. We further show that one of the identified candidates, the transcription factor MafB, controls endothelial sprouting in vitro and in vivo, and perform an integrative analysis of RNA-Seq and ChIP-Seq data to define putative direct MafB targets, which are activated or repressed by the transcriptional regulator. Together, our results identify novel cell-autonomous regulatory mechanisms controlling sprouting angiogenesis.

[1] Department of Tissue Morphogenesis, Max-Planck-Institute for Molecular Biomedicine, Röntgenstrasse 20, 48149 Münster, Germany. [2] Faculty of Medicine, University of Münster, 48149 Münster, Germany. [3] Laboratory of Regulatory Genomics, Max-Planck-Institute for Molecular Biomedicine, Röntgenstrasse 20, 48149 Münster, Germany. [4] Department of Cell and Developmental Biology, Max-Planck-Institute for Molecular Biomedicine, Röntgenstrasse 20, 48149 Münster, Germany. Correspondence and requests for materials should be addressed to J.M.V. (email: jmv@mpi-muenster.mpg.de) or to R.H.A. (email: ralf.adams@mpi-muenster.mpg.de)

The angiogenic expansion of the vascular system is indispensable for developmental growth but also contributes to disease processes in humans such as tumor formation, age-related macular degeneration or atherosclerosis[1–3]. Angiogenesis is a complex, highly coordinated multistep process in which endothelial cells (ECs) dynamically alter their behavior leading to changes at the level of differentiation, proliferation, migration, polarity, metabolism, and cell–cell communication[4–7]. Growing ECs respond strongly to a range of extracellular signals, such as matrix molecules, chemokines, growth factors, and cell adhesion molecules, which are mostly derived from the surrounding tissue, other ECs or myeloid cells. EC behavior is also tightly controlled by various transcription factors (TFs) leading to cell-autonomous changes in gene expression. Such changes are thought to integrate different external signals, but they also modulate the ability of ECs to respond to cues in their environment, which can, for example, involve the upregulation or downregulation of surface receptor expression. Even though we know many TFs that control EC specification and developmental angiogenesis[8], there is a lack of insight into the dynamic changes in endothelial gene expression during different phases of blood vessel growth, remodeling and maturation.

The formation of the retinal vasculature is a frequently used and well-characterized model system for both physiological and pathological angiogenesis in mice. Flat-mounted retinas allow the three-dimensional (3D) imaging of the local vasculature at high resolution without the limitations of tissue sectioning. In addition, vessel growth in the murine retina is initiated only after birth so that different phases of vascular development can be easily monitored and subjected to experimental manipulations[9, 10]. During the first week of postnatal life, radial outgrowth of vessels from the optic nerve head towards the peripheral retina gives rise to a superficial, two-dimensional vascular plexus. From postnatal day 7 (P7) to P15, the superficial capillaries sprout vertically to form first the deep vascular plexus in the outer plexiform layer (OPL), which is followed by the formation of the intermediate vascular plexus in the inner plexiform layer. All those three retinal vascular layers with multiple interconnecting vessels are fully formed approximately by P21[11]. Later developmental stages of the adolescent retinal vasculature are characterized by vessel pruning and remodeling processes[12, 13]. Thus, different stages of retinal vascularization represent distinct phases of angiogenesis ranging from active vascular growth to maturation and the acquisition of endothelial quiescence.

The field of transcriptome analysis using RNA sequencing platforms has developed rapidly in the past decade[14]. To date, major challenges for in vivo transcriptome analysis concern the purity of target cells, the avoidance of transcriptome-altering processing steps and the isolation of sufficient amounts of intact transcripts. The heterogeneous cellular composition of organs and the frequently tight association of different cell types complicate cell isolation and purification processes. Procedures such as organ fragmentation, manual dissection, sectioning, or enzymatic digestion combined with fluorescence-activated cell sorting or laser capture micro-dissection expose cells to a broad spectrum of physical, chemical and metabolic stresses, which are likely to affect gene expression as well as RNA quality. Circumventing such obstacles, a seminal strategy called RiboTag profiling relies on the expression of epitope-tagged ribosomal proteins, allowing the direct immunoprecipitation of ribosomes with associated (active) transcripts without time-consuming tissue digestion and cell sorting steps. In $Rpl22^{tm1.1Psam}$ RiboTag knock-in mice[15], Cre recombinase activity leads to cell type-specific and, depending on the Cre allele used in the experiment, inducible expression of a hemagglutinin (HA) epitope-tagged version of the ribosomal protein L22 (Rpl22).

In the current study, we carry out RNA-Seq analysis combined with RiboTag technology on mouse retinal ECs at five different postnatal stages—namely P6, P10, P15, P21, and P50—representing different phases of angiogenic blood vessel growth. Following differential gene expression analysis and classification of genes into seven different clusters, integrated motif activity response analysis (ISMARA)[16] leads to the identification of putative key regulators of EC behavior during retinal vascular development. One of the highest scoring candidates with no previously known role in the regulation of angiogenesis, the basic leucine-zipper (bZIP) TF MafB[17], is selected for further validation and functional characterization. In vitro assays, ChIP-Seq analysis and the characterization of EC-specific $Mafb$ mutant mice reveal that the TF promotes sprouting angiogenesis, which involves the regulation of Rho family small GTPases. The sum of our data provides new important insights into the regulation of EC behavior and the in vivo transcriptional regulation of angiogenesis.

## Results

**Profiling of retinal ECs during postnatal development**. To gain insight into dynamic changes in endothelial gene expression during different stages of retinal angiogenesis in vivo, tamoxifen-inducible and EC-specific $Pdgfb$-iCre transgenic mice[18] were interbred with $Rpl22^{tm1.1Psam}$ RiboTag knock-in mice[15] (Fig. 1a). Following tamoxifen administration on three consecutive days from P1 onwards, robust expression of HA-tagged Rpl22 protein was visible throughout the vascular endothelium of $Pdgfb$-iCre $Rpl22^{HA/HA}$ retinas at P6 and subsequent stages (Fig. 1b; Supplementary Fig. 1a). Next, transcripts were isolated by anti-HA immunprecipitation at different postnatal stages in sufficient quality and quantity for further analyses (Supplementary Fig. 1b). Quantitative RT-PCR (RT-qPCR) analysis of immunoprecipitates confirmed strong enrichment of EC-specific genes, such as $Pecam1$ and $Emcn$, while the vascular smooth muscle cell marker $Acta2$ was not enriched (Fig. 1c). Expression profiling by RNA-Seq was used for genome-wide analysis of the murine retinal EC transcriptome at P6, P10, P15, P21, and P50. These stages were chosen because they reflect key aspects of retinal angiogenesis (Fig. 1d) including outgrowth of the superficial vessel plexus (P6), vascularization of the deeper retina (P10 and P15), and vascular remodeling and maturation (P21 and P50)[10, 11]. Single sequencing libraries were generated from both retinas of each mouse and a total of 15 libraries (three biological replicates for each stage) were sequenced at an average of 8.7 M paired-end reads ($2 \times 75$ bp) resulting in 91.14% congruent paired mapped reads (Supplementary Table 1). Unsupervised hierarchical clustering and principal component analyses of all 15 RNA-Seq data sets demonstrated the high reproducibility of the obtained gene expression profiles (Fig. 1e, f; minimum replicate Pearson's correlation coefficient $\rho = 0.94$). The main source of variability in the data (principal component 1 and 2; $> 71\%$ of the variance) corresponded to time-related expression changes that reflect chronological vascular development[11]. Importantly, samples showed a consistent clustering of early (P6), mid-developmental (P10–P15), and late (P21–P50) stages of retinal angiogenesis. Taken together, these analyses establish that the experimental strategy enables the robust analysis of endothelial transcriptomes during postnatal development in vivo.

**Functionally structured transcriptional responses**. In order to study the transcriptional regulatory events behind retinal vascular development, we first identified genes with differential expression across developmental stages using Next-maSigPro, a specialized differential expression analysis for RNA-Seq time-course data[19].

From the 39,179 annotated genes in the GRCm38 mouse genome assembly, a total of 15,364 genes were regarded as differentially expressed genes (DEGs) for at least one of the time points included in this study (FDR < 0.05, $\theta = 7.33$, $R^2 = 0.5$; see

"Methods"; Supplementary Fig. 1c; Supplementary Data 1). Gene set enrichment analysis (GSEA)[20] demonstrated significant enrichment of gene sets in different developmental stages. For example, genes related to the Notch signaling pathway were

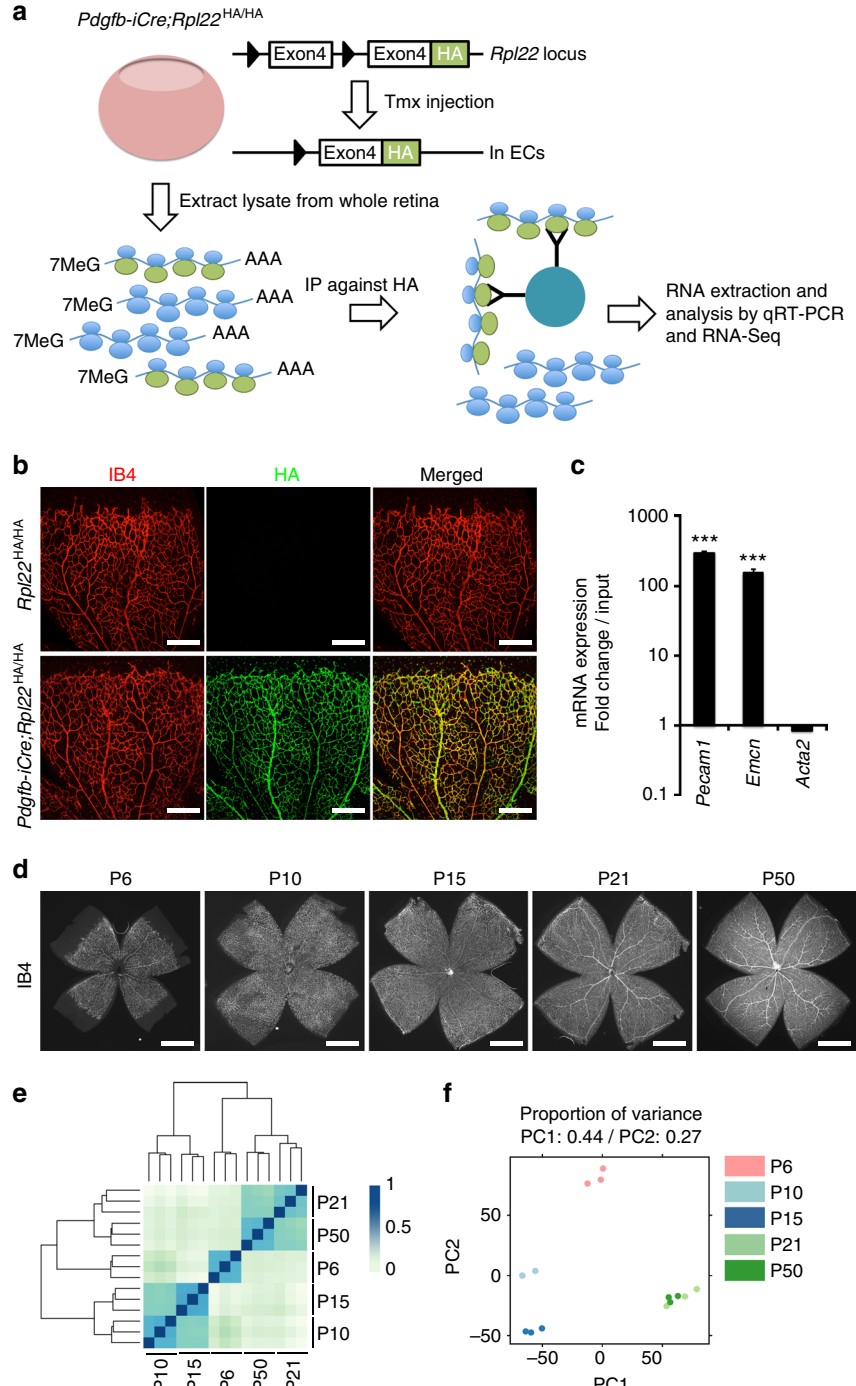

**Fig. 1** RiboTag analysis of retinal ECs during postnatal development. **a** Diagram depicting retinal EC-specific RiboTag transcriptome analysis. Inducible (Tmx controlled) recombination of the RiboTag allele (Rpl22$^{HA/HA}$) led to expression of HA-tagged L22 ribosomal protein specifically in ECs. Ribosome-bound transcripts were immunoprecipitated (IP) from homogenized whole retinas with anti-HA antibody-coupled magnetic beads. Extracted mRNAs were analyzed by RT-qPCR or RNA-Seq analysis. **b** P6 Rpl22$^{HA/HA}$ (control) and Pdgfb-iCre Rpl22$^{HA/HA}$ retinas stained with Isolectin B4 (IB4, red) and anti-HA antibody (green). Scale bars represent 300 μm. **c** RT-qPCR analysis of transcripts expressed in retinal ECs. EC-specific markers, Pecam1 and Emcn, and Acta2 negative control were normalized to Actb levels using ΔΔCT method. Anti-HA immunoprecipitated RNAs were compared to whole retinal lysate input. Error bars represent mean ± s.e.m., n = 6, ***P < 0.0001; unpaired Student's t-test. **d** IB4 staining of whole retinas at P6, P10, P15, P21, and P50 showing vascularization during postnatal development. Scale bars represent 0.5 mm. **e** Unsupervised hierarchical clustering of 15 retinal EC RNA-Seq data sets with three replicates per stage. **f** Two-dimensional principal component analysis (PCA) plot showing the discrimination of gene expression profiles between retinal ECs from different ages

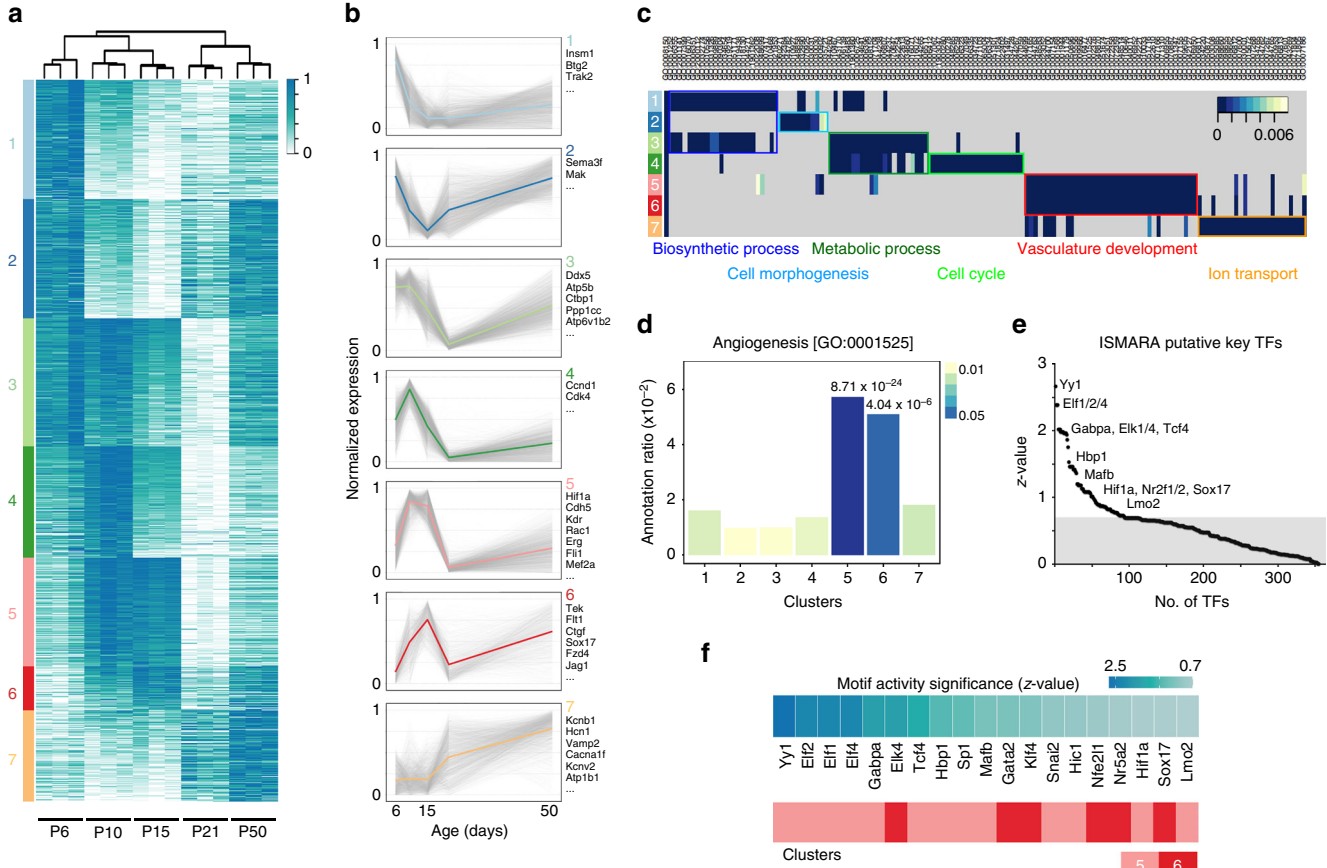

**Fig. 2** Key TFs regulating EC behaviors. **a**, **b** Normalized expression dynamics of 15,364 DEGs in retinal ECs during postnatal development are shown in a heat map (**a**) and line plots (**b**). Gene names in **b** represent typical examples within each cluster. *k*-means clustering analysis identified seven clusters of DEGs according to their expression patterns. Mean expression values of each cluster are highlighted. **c** Gene ontology (GO) terms enrichment analysis of seven DEG clusters. The most frequent and representative terms are highlighted and boxed with same colors. **d** Enrichment of genes related to Angiogenesis (GO:0001525) in each cluster. **e** ISMARA analysis of retinal EC transcriptome during postnatal development. Graph represents the significance of TF motif activity (*z*-value) with selected TFs displayed. **f** List of TFs with the highest (top 30%) scores of ISMARA analysis among genes belonging to the angiogenic cluster, clusters 5 and 6

highly enriched in early stage (P6), while JAK–STAT signaling pathway and ECM receptor interaction genes were predominantly expressed at P10 (Supplementary Fig. 2a). During the transition from mid to late developmental stages (P15 vs. P21), genes associated with the Wnt signaling pathway, tight junctions, P53 signaling pathway, and focal adhesions showed significant enrichment at P15, while genes belonging to the mitogen-activated protein kinase (MAPK) and calcium signaling pathways were enriched at P21 (Supplementary Fig. 2b).

To further investigate retinal EC DEGs, we next classified the observed transcriptional changes into main temporal expression dynamics, which give evidence for functional gene relationships and transcriptional co-regulation[21]. To this end, we divided DEGs into seven clusters with characteristic expression patterns and dynamics by applying the *k*-means clustering algorithm with the Elbow method (see "Methods") (Fig. 2a, b; Supplementary Fig. 2c, d). Each cluster is distinctively composed of genes with particular biological functions, as highlighted by the cluster-specific enrichment of known functional gene sets such as gene ontology terms (Fig. 2c; Supplementary Fig. 3a) or KEGG/Reactome signaling pathway members (Supplementary Fig. 3b), all of which suggest the functional relevance of the structured transcriptional response.

Genes that encode proteins for the biosynthesis of RNA, macromolecules, nucleobase-containing compounds, and aromatic and heterocyclic compounds are highly enriched in clusters

1 and 3, representing genes that are expressed abundantly at the early stage of postnatal development (Fig. 2c). Dual expression peaks at P6 and P50 (cluster 2) were observed for genes required for morphogenesis, assembly and organization of cilia and other cell projections, and cell morphogenesis. Clusters 3 and 4, in which genes showed peak expression at P10, were enriched in genes related to metabolism of macromolecules, nucleobase-containing compounds, and aromatic and heterocyclic compounds. Genes regulating the mitotic cell cycle were enriched in cluster 4 and genes regulating ion transmembrane transport were concentrated in cluster 7, which represents genes that show increased expression at later developmental stages, namely P21 and P50. Interestingly, genes regulating blood vessel development, vasculature development, and angiogenesis were expressed at the highest levels during P10 and P15, and were greatly enriched in clusters 5 and 6 (Fig. 2c, d). Moreover, these two clusters included genes that encode proteins for signal transduction, cell communication, response to stimuli, and cell surface receptor signaling pathways, suggesting a major contribution of clusters 5 and 6 genes in the control of retinal vascular development. Based on these findings, we focused our subsequent analysis on the 3248 genes contained in the "angiogenic" clusters 5 and 6.

To identify the transcriptional regulators of the observed endothelial gene expression dynamics, genome-wide promoter activity was analyzed with ISMARA, a web-based automated tool

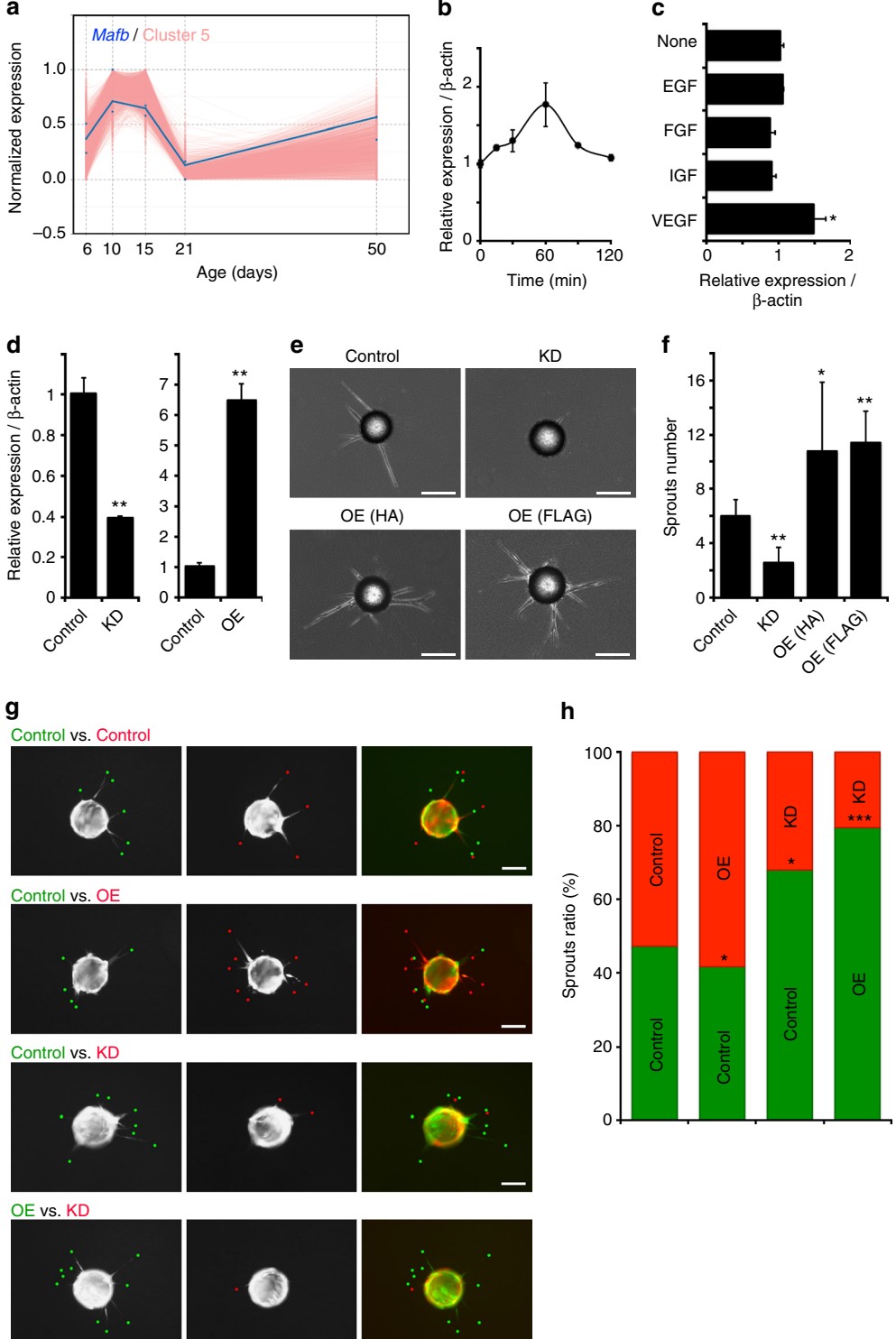

**Fig. 3** MafB is a novel TF regulating sprouting angiogenesis in vitro. **a** Normalized expression of *Mafb* (*blue*) among cluster 5 genes (*pink*) in postnatal retinal ECs. **b**, **c** RT-qPCR analysis of relative *Mafb* expression in MS1 ECs after treatment of complete EGM-2 media (**b**) or single growth factors, as indicated (**c**). n = 4, *P < 0.05; unpaired Student's t-test. *Error bars* represent mean ± s.e.m. **d** RT-qPCR analysis of relative *Mafb* expression in MS1 EC lines infected with lentivirus containing no-insert (Control), *Mafb*-targeting shRNA (KD, knockdown) or HA-tagged *Mafb* cDNA (OE, overexpression). n = 3, **P < 0.001; unpaired Student's t-test. *Error bars* represent mean + s.e.m. **e** Three-dimensional fibrin gel bead sprouting assay with stable MS1 lines expressing no insert (Control), *Mafb*-targeting shRNA (KD), HA-tagged *Mafb* cDNA (OE (HA)), or FLAG-tagged *Mafb* cDNA (OE (FLAG)). Representative spheroids are shown for each condition at culture day 3. *Scale bars* represent 170 μm. **f** Average number of sprouts per spheroid in each condition of **e**. n = 12, *Error bars* represent mean + s.d. *P < 0.05, **P < 0.001; unpaired Student's t-test. **g** Competitive sprouting assay with 1:1 mixture of differentially labeled (eGFP or tdTomato) MS1 lines expressing no insert (Control), *Mafb*-targeting shRNA (KD) or HA-tagged *Mafb* cDNA (OE). Representative spheroids are shown for each condition at 8 h after the fibroblast seeding. *Scale bars* represent 100 μm. **h** Ratio of differentially labeled sprouts in each condition shown in **g**. n = 12, *P < 0.05, ***P < 0.0001; unpaired Student's t-test

for the computational reconstruction of regulatory circuitry[16]. Applying ISMARA to our retinal EC transcriptome data set, we found 93 TFs as putative key regulators according to their significant motif activities across the developmental stages (z-value > 0.7, Fig. 2e and Supplementary Data 2). Among the top-ranked TF motifs with the best explanatory power identified by ISMARA, already known key angiogenic factors such as Hif1α, Sox17, Lmo2, Elf1/2/4, and Nr2f2 were included. In addition, novel transcriptional regulators of vascular development, such as

Yin yang 1 (Yy1), the ETS family member GABPalpha (Gabpa), or the bZIP TF MafB, were also predicted. By combining the results of DEG clustering and ISMARA prediction, we chose MafB for further functional characterization because the TF was found among the top-ranked candidates based on ISMARA prediction (top 30%), was included in the angiogenic cluster (Fig. 2f), and was upregulated in retinal ECs during early postnatal development (Fig. 3a).

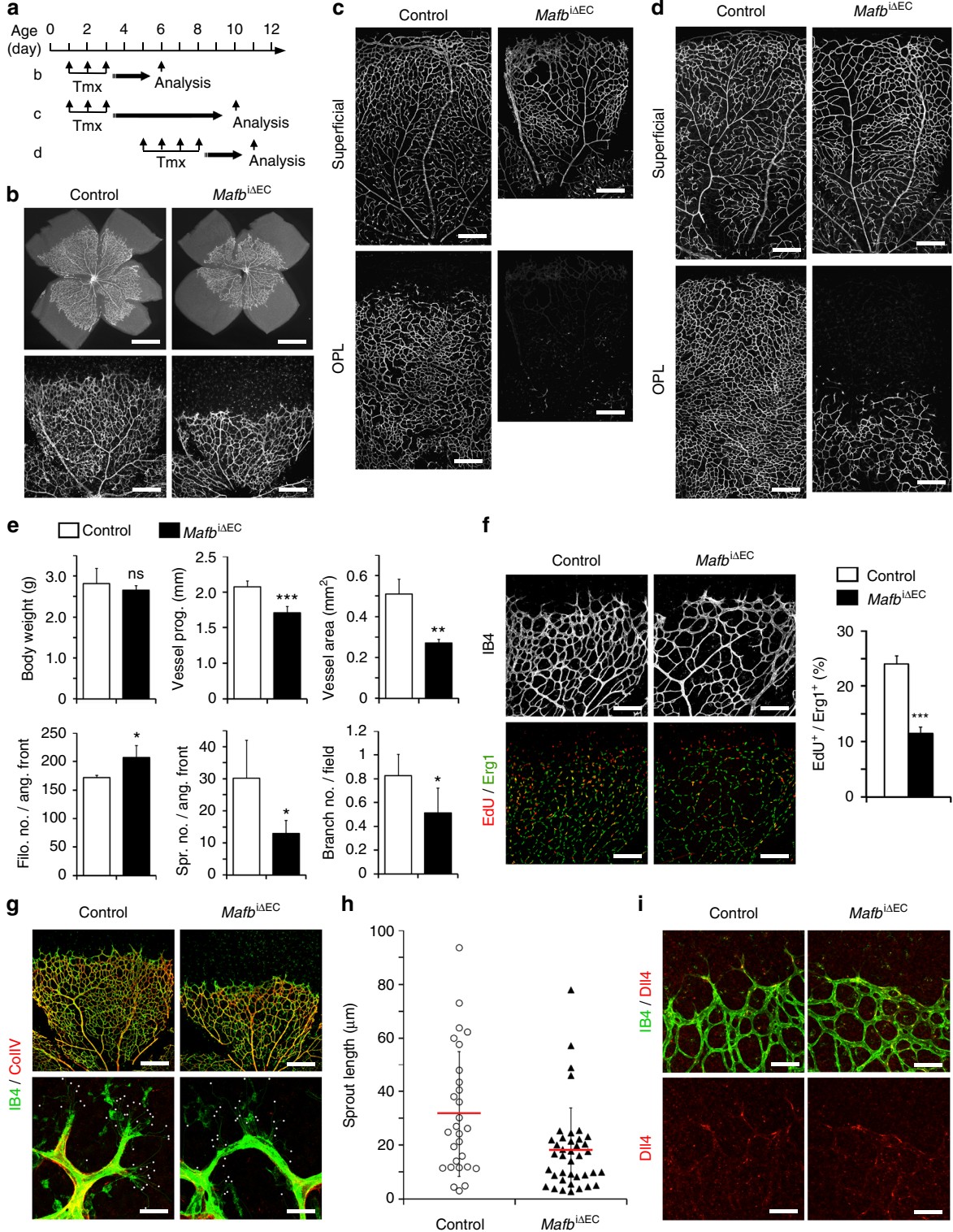

**MafB is indispensable for endothelial sprouting in vitro.** As a member of the large Maf TF family, MafB regulates tissue-specific gene expression and cellular differentiation in various organs including brain, kidney, pancreas, skin, and blood[22–25]. Recently, MafB (mafba in zebrafish) has been implicated in the regulation of lymphangiogenesis but expression, although at lower level, was also detected in embryonic blood vessels and in cultured blood vessel ECs[26, 27]. First, we addressed whether extracellular stimulation induces *Mafb* expression in ECs in vitro. After 24-h starvation, we treated mouse vascular MS1 ECs (see "Methods") with complete EGM-2 media or single growth factors and analyzed isolated total RNA at various time points from 0 to 120 min using RT-qPCR. Following EGM-2 stimulation, *Mafb* expression was significantly upregulated at 1 h after stimulation and dropped back to baseline at later time points (Fig. 3b). Increased *Mafb* expression was also observed after stimulation with VEGF-A but not with several other growth factors (Fig. 3c).

Next, we used a lentiviral infection system to establish mouse EC lines in which MafB is stably overexpressed or its endogenous expression suppressed (Fig. 3d; Supplementary Fig. 4a). EC sprouting from beads in 3D fibrin gels was strongly reduced after *Mafb* knockdown, whereas overexpression of epitope-tagged MafB led to significantly increased sprouting (Fig. 3e, f). Competitive sprouting assays, in which differentially labeled ECs had been mixed in a 1:1 ratio, showed that MafB expression positively correlated with the number of endothelial sprouts (Fig. 3g, h). *Mafb* knockdown also strongly impaired directional migration and collective migration in vitro and led to reduced adhesion to Collagen I-coated surfaces (Supplementary Fig. 4b±d). Taken together the results above show that MafB is controlled by pro-angiogenic stimuli and promotes angiogenic EC behavior in a cell-autonomous fashion in vitro.

**Defective retinal angiogenesis in EC-specific *Mafb* knockout mice.** We next determined MafB expression in the developing retina by immunostaining of *Cdh5-EGFP* transgenic reporter mice, which express membrane Tomato and nuclear EGFP specifically in ECs. Nuclear MafB protein was observed in multiple cell types at different levels (Supplementary Fig. 5a). In the vascular endothelium, MafB was enriched in ECs at the distal end of sprouts, which is consistent with the VEGF-mediated induction of the gene (Supplementary Fig. 5b, c). MafB was also visible in the nuclei of arterial ECs (Supplementary Fig. 5d).

To investigate the function of MafB in the endothelium during physiological angiogenesis in vivo, we generated EC-specific inducible *Mafb* KO mice (*Pdgfb-iCre Mafb*[p/p] or *Mafb*[iΔEC]) (Supplementary Fig. 6a, b). Cre-mediated inactivation of *Mafb* was induced by daily intraperitoneal injection of tamoxifen at P1–P3 for early induction, or at P5–P8 for late induction, as described previously[10]. Retinal blood vessels were analyzed at P6 or P10 following early induction, or at P11 following late induction (Fig. 4a). Consistent with the in vitro sprouting assay

results, inactivation of *Mafb* in ECs led to markedly reduced outgrowth of the superficial retinal vessel plexus at P6 (Fig. 4b). Defects were even more pronounced at P10 when sprouting into the deeper retina was strongly impaired (Fig. 4c; Supplementary Fig. 6c). *Mafb* inactivation at P11 did not affect the superficial vessel plexus, which had formed before the administration of tamoxifen, but disrupted the vascularization of OPL in the retina (Fig. 4d). In contrast, *Mafb* inactivation at later stages of development (P10–P13 or P21–P25) did not result in overt defects suggesting that the TF is not required for vessel integrity and remodeling. These results establish that MafB is essential for retinal angiogenesis in early postnatal life, which is consistent with its expression pattern in ECs (Fig. 3a).

Detailed characterization of the P6 *Mafb*[iΔEC] retinal vasculature confirmed that EC sprouting, vascular area, and vessel branch formation were significantly reduced relative to control littermates, while body weight was comparable in both groups (Fig. 4e). Administration of 5-ethynyl-2′-deoxyuridine (EdU) for 2 h prior to analysis showed decreased proliferation of *Mafb*[iΔEC] ECs (Fig. 4f). Normal Dll4 expression at the *Mafb*[iΔEC] vascular growth front and a slightly but significantly increased number of filopodia suggested that MafB is not essential for the specification of endothelial tip cells (Fig. 4e, g, i), whereas the length of mutant sprouts was strongly decreased (Fig. 4h). In addition, loss of MafB in ECs led to increased vascular regression and decreased mural cell coverage, indicative of defects in blood vessel vascular stability and maturation, respectively (Supplementary Fig. 6d, e). Collectively, these results show that MafB is an important regulator of EC behavior that controls vascular sprouting and multiple other aspects of angiogenesis in vivo.

**Gene expression changes in MafB-deficient retinal ECs.** MafB can both activate or inhibit gene expression in a cell context-dependent fashion[28]. To identify MafB-regulated genes in the postnatal endothelium, we performed a combined RiboTag RNA-Seq analysis of P6 retinal ECs from *Pdgfb-iCre Mafb*[p/p] *Rpl22*[HA/HA] mutants (i.e., *Mafb*[iΔEC] mutants in the *Rpl22*[HA/HA] background) and *Pdgfb-iCre Mafb*[wt/wt] *Rpl22*[HA/HA] controls. As expected, *Mafb* expression was markedly reduced in *Mafb*[iΔEC] heterozygotes and homozygotes by RT-qPCR or RNA-Seq analyses (Fig. 5a; Supplementary Data 3). RNA-Seq analysis of three biological replicates showed 1240 downregulated genes and 1129 upregulated genes after *Mafb* inactivation (Fig. 5b). Downregulated genes, representing targets that are positively regulated by MafB, were enriched for GO terms associated with developmental processes, while upregulated genes, representing negative targets, were enriched for terms associated with mitosis, cell cycle regulation and metabolism (Fig. 5c). Mammalian phenotype enrichment analysis (MamPhEA)[29] further demonstrated that MafB-regulated genes in retinal ECs were associated with phenotypes such as abnormal vasculature morphology, abnormal vascular EC physiology, abnormal apoptosis, abnormal

**Fig. 4** Retinal angiogenesis in EC-specific MafB KO mice. **a** Diagram of *Mafb* inactivation and mutant analysis. Tmx was injected at P1–P3 for early induction in **b** and **c** or at P5–P8 for late induction in **d**. Mice were sacrificed and analyzed at indicated ages. **b–d** IB4 staining of retinas from *Mafb*[iΔEC] and control littermates (Control) at P6 (**b**) or P10 (**c**) following the early induction, and at P11 following the late induction (**d**). *Scale bars* represent 0.5 mm (**b** upper) and 300 μm (**b** lower, **c** and **d**). **e** Analysis of control or *Mafb*[iΔEC] retinal vasculature. The number of EC filopodia and sprouts were counted and normalized for 1 mm in length of vessels at the angiogenic front. The number of branching points were defined as intersections between vessels within a field of 1000 μm². $n = 6$, *Error bars* represent mean + s.d. *$P < 0.05$, **$P < 0.001$, ***$P < 0.0001$; unpaired Student's $t$-test; ns, not significant. **f** P6 control or *Mafb*[iΔEC] retinas stained with IB4 (*white*), EdU (*red*), and anti-Erg1 antibody (*green*). *Scale bars* represent 150 μm. Graph shows the ratio of EdU⁺ ECs in Control and *Mafb*[iΔEC] retinas at P6. $n = 6$, *Error bars* represent mean + s.d., ***$P < 0.0001$; unpaired Student's $t$-test. **g** P6 control or *Mafb*[iΔEC] retinas stained with IB4 (*green*) and anti-collagen IV (*CollV; red*). *White dots* indicate filopodia. *Scale bars* represent 300 μm (*upper row*) and 25 μm (*lower row*). **h** Length of sprouts in P6 control or *Mafb*[iΔEC] retinas. $n = 6$, *Error bars* represent mean (*red line*) ± s.d., $P < 0.0001$; unpaired Student's $t$-test. **i** P6 control or *Mafb*[iΔEC] retinas stained with IB4 (*green*) and anti-Dll4 (*red*). *Scale bars* represent 75 μm

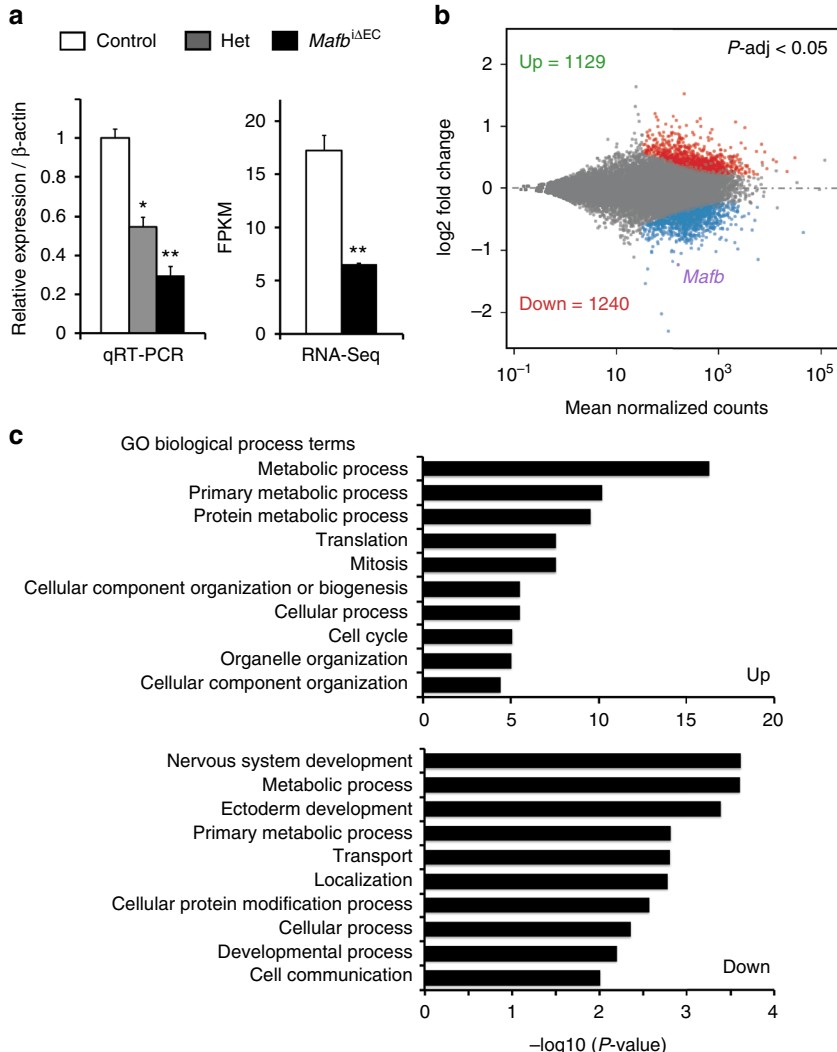

**Fig. 5** RiboTag/RNA-Seq analysis of EC-specific *Mafb* mutant mice. **a** *Mafb* expression in P6 control, *Mafb* heterozygous mutant (*Pdgfb-iCre Mafb*[+/P]; Het) or *Mafb*[iΔEC] retinal ECs assessed by RT-qPCR or RNA-Seq. $n = 3$, *Error bars* represent mean + s.e.m. *$P < 0.05$, **$P < 0.001$; unpaired Student's *t*-test. **b** MA plot of DEGs between P6 *Mafb*[iΔEC] ECs and control retinal ECs. $n = 3$, *red dots*, upregulated genes (1129); *blue dots*, downregulated genes (1240); *purple dot*, *Mafb*. **c** Top GO biological process terms enriched in up- or downregulated genes in *Mafb* mutant retinal ECs at P6

placenta morphology, and abnormal nervous system morphology (Supplementary Fig. 7). Because the vascular and the nervous system show similarities at the morphological and molecular levels, and share common developmental principles coordinating their physiological growth[30], these results support the important cell-autonomous role of MafB in ECs seen in vivo and in vitro.

**ChIP-Seq analysis of MafB in ECs**. Next, we addressed which of the gene expression changes observed in *Mafb*[iΔEC] ECs in vivo can be directly attributed to MafB. To this end, we performed ChIP-Seq analysis of cultured MS1 ECs stably overexpressing HA-tagged MafB. This led to the identification of 2438 peaks associated with MafB binding in ECs (Fig. 6a), which were distributed across the whole genome but enriched in genic regions, especially in the promoter and first intron (Fig. 6b). De novo motif discovery using HOMER software[31] identified a MafB motif that resembled both the canonical Maf-recognition element (MARE; GCTGA(G/C)TCA(T/C)) and a slightly expanded version of the MafB motif found in the JASPAR database (Fig. 6c). Interestingly, genomic regions enrichment of annotations tool (GREAT) analysis[32] demonstrated that genes associated with MafB peaks are highly enriched for relevant phenotype terms,

such as abnormal vascular EC physiology and abnormal induced retinal neovascularization. Consistently, disease ontology terms for human vasculature diseases, such as aneurysm, retinal vascular disease, diabetic retinopathy and diabetic angiopathy, and carotid artery disease were also significantly enriched in MafB peak-associated genes (Fig. 6d). Moreover, MafB peak-associated genes were also strongly enriched in the angiogenic clusters 5 and 6 (Fig. 6e) suggesting that MafB directly regulates the expression of genes controlling vascular growth and integrity.

**MafB-controlled target genes regulate EC behavior**. Next, we identified direct and functional target genes of MafB by intersecting genes regulated by MafB in retinal ECs with MafB-bound genes identified in the ChIP-Seq analysis. We observed significant overlap of those two different gene sets (Fisher's exact test, $P$ value $< 2.2 \times 10^{-16}$), with 307 genes identified as putative direct targets of MafB (Fig. 7a). These genes were distributed in all seven clusters but were significantly enriched in the angiogenic cluster 5 relative to all DEGs in retinal ECs (Fig. 7b; Fisher's exact test, $P < 3.43 \times 10^{-8}$). In particular, modulators of G-proteins (PANTHER Protein Class PC00022) were significantly enriched among the direct targets of MafB (Bonferroni correction for multiple testing,

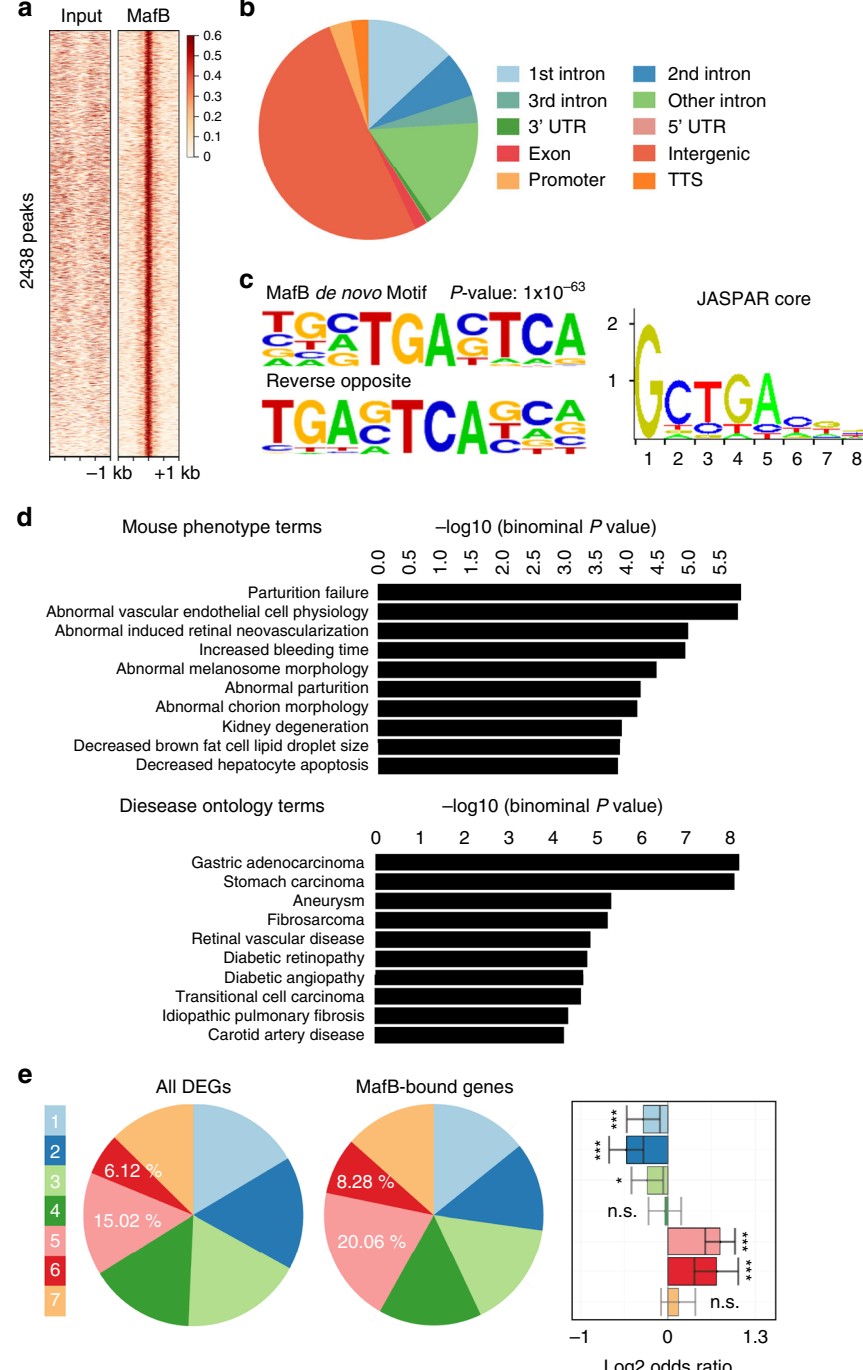

**Fig. 6** ChIP-Seq analysis of MafB in ECs. **a, b** Heat map of 2438 peaks of MafB ChIP-Seq with input (**a**) and distribution of MafB peaks across the genome (**b**). *Color scale* corresponds to reads per million mapped reads in 20 bp bins. **c** MafB de novo motif identified by HOMER with JASPAR MafB (*Rattus norvegicus*) motif. **d** Enriched mouse phenotype terms (*top*) and disease ontology terms (*bottom*) for genes associated with MafB in the context of ECs. **e** Association between MafB-bound genes and retinal clusters of differential gene expression (Fisher's exact test with FDR adjustment for multiple testing; ns, not significant; *q-value < 0.05; **q-value < 0.01; ***q-value < 0.001)

$P = 0.0261$), suggesting that impaired G-protein signaling might be linked to the defective sprouting morphogenesis of $Mafb^{i\Delta EC}$ ECs in vivo. G-protein-coupled receptor-2-interacting protein-1 (Git1) and Rho GDP dissociation inhibitor-beta (Arhgdib, also known as RhoGDI2) were selected as the most promising MafB-effector target genes regulating G-protein signaling and EC behavior. Recent studies have shown that Git1 is important for the directional migration of ECs and sprouting angiogenesis during lung development or tumor growth in vivo[33, 34]. Git1 activates Rac1/Cdc42 and Mek1–Erk1/2 pathways to induce

cortactin activation and lamellipodia formation, which is critical for the directional migration of ECs[33]. In contrast, Arhgdib/RhoGDI2 is known to inhibit Rho GTPases, preferentially Rac1 and Cdc42, by direct interaction and thereby maintains Rho proteins in the inactive, GDP-bound state[35]. Binding of MafB was detected in the first intron of the *Git1* and *Arhgdib* genes (Fig. 7c). In MafB-deficient retinal ECs, *Git1* transcript expression was decreased, whereas *Arhgdib* mRNA was increased, which was confirmed by RT-qPCR analysis (Fig. 7d, e). It has been reported that MafB has two modes of action for the regulation of target

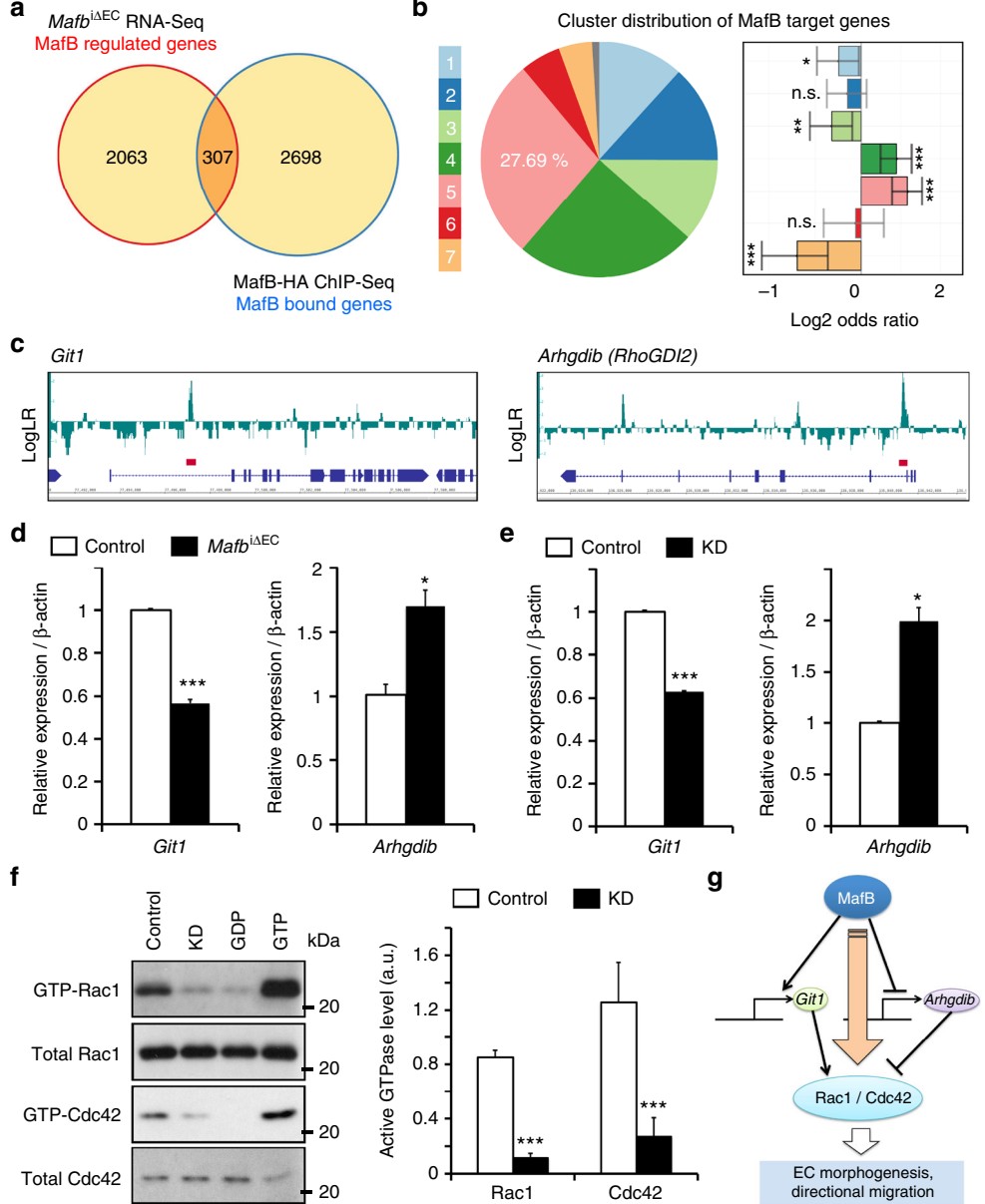

**Fig. 7** Direct target genes of MafB for the regulation of EC behavior. **a** Venn diagram of putative MafB-regulated genes identified by P6 retinal EC RNA-Seq and MafB ChIP-Seq. **b** Association between MafB target genes and retinal clusters of differential gene expression. Only genes overlapping in **a** are used for this analysis. (Fisher's exact test with FDR adjustment for multiple testing; ns, not significant; *q-value < 0.05; **q-value < 0.01; ***q-value < 0.001). **c** MafB ChIP-Seq tracks of *Git1* and *Arhgdib* genes; *red bars*, MafB peaks called by HOMER. **d**, **e** RT-qPCR analysis of *Git1* and *Arhgdib* in P6 retinal ECs (**d**) or MS1 mouse EC lines (**e**). n = 3, *P < 0.05, ***P < 0.0001; unpaired Student's t-test, *Error bars* represent mean + s.e.m. **f** Precipitation and western blot detection of active, GTP-bound Rac1 and Cdc42 in *Mafb* knockdown (KD) MS1 ECs in vitro. Graph shows quantification by densitometric analysis. Active Rac1/Cdc42 levels were normalized with the total Rac1/Cdc42 in whole cell lysates. GDP and GTP demonstrate negative and positive controls of active Rac1/Cdc42 levels, respectively. n = 3, *Error bars* represent mean + s.d., ***P < 0.0001; unpaired Student's t-test. **g** Schematic model outlining the transcriptional regulation of EC behavior by MafB. MafB regulates expression of *Git1* and *Arhgdib* simultaneously but differentially, which facilitates Rac1 and Cdc42 activation and thereby promotes vascular morphogenesis and directional EC migration

gene expression, which are MARE-dependent or MARE-independent[36]. Interestingly, the first intron of *Git1* contains a MafB-binding motif whereas RhoGDI2 lacks a MARE consensus sequence, suggesting distinct mechanisms of MafB target gene regulation in ECs. Knockdown of *Mafb* expression in MS1 ECs led to strongly reduced levels of active Rac1 and Cdc42 (Fig. 7f, g), suggesting that MafB-dependent upregulation of Git1 and downregulation of RhoGDI2 control the activity of Rho family GTPase synergistically. Rac1[37, 38] and Cdc42[39, 40] are critical regulators of endothelial sprouting and control central aspects of

EC behavior including lamellipodia formation, adhesion and migration. Consistent with the observed downregulation of active Rac1 and Cdc42, MafB-depleted ECs showed significantly reduced directional migration in a 2D VEGF-A gradient as well as impaired collective migration and adhesion (Supplementary Fig. 4b–d). To determine whether re-activated small GTPase activity can rescue the MafB KD phenotype, we transfected constitutively active form of Rac1 (Rac1-V12) into control (scrambled siRNA-treated) or *Mafb* KD MS1 ECs (Supplementary Fig. 8a, b). As a result, the collective migration and sprouting

defects in *Mafb* KD cells were rescued (Supplementary Fig. 8c, d). Taken together, these results identify MafB as a key regulator of EC migratory behavior in response to external signals, which involves both the upregulation and suppression of target genes (Fig. 7g).

## Discussion

Gene expression profiling is a powerful approach for the analysis of cellular behavior and the characterization of molecular mechanisms controlling a wide range of biological processes. In the case of angiogenesis, numerous in vitro studies have explored how gene expression in ECs is altered after exposure to stimuli such as pro-angiogenic signals, or during the formation of tubular structures in 3D culture systems[41–45]. While these studies have provided valuable insights, cultured ECs can only recapitulate certain aspects of angiogenic vessel growth, lack exposure to crucial flow and tissue-derived signals, and fail to undergo differentiation, remodeling and maturation processes, which are crucial for the formation of a fully functional vascular network in vivo. By contrast, studies using ECs from in vivo models of angiogenesis typically struggle with issues such as the amount and purity of the freshly isolated cells, and are complicated by time-consuming and presumably expression-altering tissue dissociation and cell sorting steps. The latter are avoided by Ribosome/ RiboTag profiling, which, in combination with cell type-specific genetic approaches, enables the direct and rapid immunoprecipitation of ribosome-bound transcripts at high purity. This approach has been used, for example, for the characterization of gene expression in different central nervous system (CNS) cell types[46, 47], of CNS astrocytes during scar formation and regeneration, or of mRNA translation in axons of developing and adult neurons[48]. These studies have also greatly benefitted by recent advancements in RNA-Seq technology and in the bioinformatic analysis of large transcript data sets[49].

In the current study, the integration of such methods has enabled the generation of an extensive data set of EC gene expression during different, biologically defined key stages of retinal angiogenesis. This has led to the in silico identification of seven gene expression clusters reflecting different functions and signaling pathway activities. While early stages of retinal angiogenesis are characterized by transiently high levels of genes involved in biosynthetic processes, metabolism and cell cycle regulation, many of the well-known regulators of angiogenesis, including critical cell adhesion molecules, receptors, ligands, and TFs, are enriched in clusters 5 and 6. Later stages of vascular morphogenesis in the retina show a sharp downregulation of these genes suggesting that this reduced expression of pro-angiogenic regulators might enable vessel stabilization, remodeling and maturation steps. However, the upregulation of anti-angiogenic factors or changes in endothelial metabolism, as proposed previously[50, 51], or ion homeostasis may also contribute to the transition from active growth to quiescence. We propose that similar genes and expression clusters might prove relevant for angiogenesis in other organs. Furthermore, the deregulation of these genes, such as abnormal and persisting upregulation of biosynthetic, metabolic and pro-angiogenic regulators may be relevant for processes of pathological angiogenesis, which are often characterized by deregulated growth, lack of vessel maturation and defective remodeling[52–54].

Many TFs control processes such as EC proliferation, junction formation or arteriovenous differentiation, and thereby play critical roles in normal and pathological blood vessel growth. For example, Ets family TFs (Elf1/2/4 and Erg), which are among the highest scoring TFs in our analysis, are major regulators of EC function and angiogenic growth[55, 56]. They have been also shown

to act in combination with Forkhead TFs, which are crucial for vascular morphogenesis[50, 57]. Another transcriptional regulator identified in our analysis is one of the most important known regulators of arteriovenous differentiation, namely Nr2f2 (COUP-TFII), an orphan nuclear receptor that is expressed by venous ECs[58, 59]. Among the TFs that have not been previously associated with blood vessel morphogenesis, we have focused on MafB, a member of the Maf TF family and the large AP-1 superfamily[60]. Global inactivation of murine *Mafb* led to perinatal lethality and showed that the gene is required for several developmental processes such as patterning of the embryonic hindbrain, podocyte differentiation and survival of renal tubule epithelial cells, aspects of brain stem neurogenesis, and islet beta cell maturation in the pancreas[25, 61–63]. More recently, MafB was also shown to maintain the mature status of lymphatic ECs by inducing expression of Prox1 and other TFs controlling lymphatic vessel development[64]. In the zebrafish embryo, MafB controls the migration of lymphatic EC precursors emerging from the posterior cardinal vein[27]. Both studies also reported that MafB expression is induced by signaling interactions between VEGF-C and its main receptor, the tyrosine kinase VEGFR3. The latter is consistent with our own data indicating that *Mafb* expression can be also induced by VEGF-A and, presumably, via activation of the receptor VEGFR2 (Fig. 3b, c). The new findings also establish that MafB is required for angiogenesis in vitro and in vivo, which involves the regulation of small GTPases controlling cytoskeletal dynamics and thereby cell migration (Fig. 7g). This link between MafB and cytoskeletal regulators may well prove relevant for lymphangiogenesis and other biological functions of the TF.

Interestingly, the role of MafB in ECs is not confined to the induction of gene expression but also appears to be mediated by target repression of targets. MafB was previously shown to control erythroid development by repressing Ets-1-mediated gene expression, which, however, involves direct binding of the two TFs[65] rather than the formation of a DNA-bound repressor complex. During hindbrain segmentation in the early embryo, MafB was found to induce expression of the TF Hoxa3 but repressed the expression of Hoxb1[61]. Activating and repressing roles for MafB and the related Maf/MafA protein were also observed during kinetic transcriptome analysis of epidermal regeneration in vitro[22]. Whether MafB-dependent gene downregulation in these studies was mediated directly, i.e., via formation of DNA-bound repressor complexes, through protein-protein interactions (as shown for Ets-1), or through altered expression of downstream TFs remains to be addressed.

Taken together, our study has generated a detailed EC-specific in vivo gene expression atlas that enables the prediction of gene function in critical steps or developmental stages of angiogenic vessel growth. As shown for the example of MafB, this powerful approach enables the identification of novel regulators and molecular processes mediating angiogenic growth of blood vessels.

## Methods

**Mice and inducible genetic modifications**. We generated loxP-flanked *Mafb* conditional KO mice (*Mafb*[p/p], Supplementary Fig. 6b). *Pdgfb-iCre* transgenic mice were bred into a background of RiboTag (*Rpl22*[HA/HA]), and/or *Mafb*[p/p] mice. To induce Cre-mediated gene recombination, tamoxifen (Tmx; T5648; Sigma-Aldrich) was dissolved in 100% ethanol, further diluted with peanut oil, and then intra-peritoneally injected at P1–P3 for early induction (50 μl of 1 mg/ml Tmx per day) or at P5–P8 for late induction (50 μl of 2 mg/ml Tmx per day), respectively. Tmx-injected *Mafb*[p/p] littermates were always used as controls. *Cdh5-EGFP* transgenic mice were generated by pronuclear injection into fertilized mouse oocytes. The injected construct consists of a cDNA encoding membrane-tagged tdTomato (Addgene plasmid # 17787), 2A peptide, AU1 tag, and H2B-EGFP (Addgene plasmid # 11680) introduced by homologous recombination into the start codon of

*Cdh5* in PAC clone 353-G15. All mice were maintained on the C57BL/6 background.

All animal experiments were performed in compliance with the relevant laws and institutional guidelines, approved by local animal ethics committees, and conducted with permissions granted by the Landesamt für Natur, Umwelt und Verbraucherschutz (LANUV) of North Rhine-Westphalia, Germany.

**Mouse retinal vasculature analysis.** All immunostainings and quantitative analysis of mouse retina were performed as described previously[10]. Both male and female mice of the indicated age were used. In brief, whole animal eyes were fixed in 4% paraformaldehyde (PFA, P6148; Sigma Aldrich) at 4 °C for 2 h (for multiple whole-mount immunohistochemistry) or overnight (for quantitation of retinal vasculatures and filopodia/sprout analysis). After PBS washing, retinas were dissected and partially cut in four quadrants. Blocking/permeabilization with 1% BSA and 0.3% Triton X-100 in PBS for 30 min, 3× washing with Pblec buffer (1% Triton X-100, 1 mM CaCl$_2$, 1 mM MgCl$_2$ and 0.1 mM MnCl$_2$ in PBS), and incubation with Pblec buffer containing primary antibodies (1:50 dilution) at 4 °C for overnight were followed by secondary antibody staining using suitable species-specific Alexa Fluor-coupled antibodies (Invitrogen, 1:200 dilution) and flat-mounting in microscope glass slides with Fluoromount-G (0100-01; SouthernBiotech). Can Get Signal immunostain solution B (Toyobo) was used to enhance the intensity and specificity of immunostaining. We performed immunostaining using following antibodies: biotinylated IB4 (B-1205; VectorLabs), rat anti-HA (3F10; Roche), rabbit anti-MafB (IHC-00351-1; Bethyl Laboratories), rabbit anti-Erg1 (EPR3863; Abcam), rabbit anti-ColIV (2150-1470; Serotec), goat anti-Dll4 (AF1389; R&D), goat anti-Pdgfrb (AF1042; R&D), and streptavidin Alexa Fluor-488 (Invitrogen). For labeling of proliferating cells, 50 µl of 10 mM EdU (Life Technologies) was injected intraperitoneally 2 h before sacrifice. Mouse retinas were processed as described and EdU-positive cells were detected with the Click-iT EdU Alexa Fluor-647 Imaging Kit (C10340; Life Technologies). Quantifications of retinal vasculature were performed on high-resolution confocal images of at least six retina samples per group with the Volocity software (Perkin Elmer).

**Isolation of polysome-bound mRNAs using RiboTag.** RiboTag analysis was performed as described previously with minor modifications[15]. Both male and female mice of the indicated age were used. Mouse retinas were isolated, immersed in liquid nitrogen immediately to snap freeze, and stored at −80 °C for later use. We added 400 µl of polysome buffer (50 mM Tris, pH 7.5, 100 mM KCl, 12 mM MgCl$_2$, 1% Nonidet P-40, 1 mM DTT, 200 U/ml RNasin, 1 mg/ml heparin, 100 µg/ml cyclohexamide, and 1× protease inhibitor mixture) to each sample and disrupted the tissue in 1.5 ml microcentrifuge tubes using pellet pestles (749540-0000; Kimble Chase). Postmitochondrial supernatant was formed by centrifugation for 10 min at 12,000 rpm at 4 °C and 10 µl of the supernatant were saved for input. For immunoprecipitation (IP) against HA, we added 25 µl of anti-HA antibody-conjugated magnetic beads (M180-11; MBL) to the retinal supernatant prior to incubation on an orbital shaker at 4 °C overnight. IP beads were washed for four times with high-salt buffer (50 mM Tris, pH 7.5, 300 mM KCl, 12 mM MgCl$_2$, 1% Nonidet P-40, 1 mM DTT, and 100 µg/ml cyclohexamide) and resuspended in 350 µl of RLT plus buffer plus β-mercaptoethanol. Total RNAs were extracted using RNeasy Micro kit (Qiagen) according to manufacturer's instructions. The quality and quantity of the RNA samples were analyzed using Bioanalyzer with RNA 6000 pico kit (Agilent).

**RNA-Seq and gene expression analysis.** RNA library construction was performed with the Illumina TruSeq Stranded Total RNA Sample Preparation Kit with Ribo-Zero (Illumina) according to the manufacturer's instructions. The resulting mRNA library was sequenced on a MiSeq sequencer using 2 × 75 bp paired-end MiSeq v3 chemistry (Illumina). Sequenced reads were aligned to the mouse (mm10) reference genome with TopHat (version 2.0.12)[66], and the aligned reads were used to quantify mRNA expression by using HTSeq-count (version 0.6.1)[67]. DESeq2[68] was used to identify DEGs across the samples. To identify genes with significant expression profile differences over time between retinal developmental stages, we used Next-maSigPro[19], an R Bioconductor package that works by fitting a generalized linear model with negative binomial dispersion to the library-corrected gene reads counts, but explicitly accounting for the time-dependence of the polynomial model. The method regards a gene as DE if the coefficients of its fitted model are significant for any time variable. Both, the correction for RNA-Seq library-specific artifacts [trimmed mean of *M*-values (TMM) normalization] and the estimation of the θ dispersion parameter were done using the implemented R Bioconductor edgeR package[69]. The TMM normalization factors were always close to 1. After benchmark analyses, next-maSigPro was run with the parameters: θ = 1/(edgeR Φ) = 7.55; FDR < 0.05; $R^2$ = 0.5. The different stages of the DGE analysis are visualized in Supplementary Fig. 1c.

**Clustering of gene expression dynamics and other analyses.** For each gene, the TMM normalized gene counts where linearly scaled to a 0–1 range to minimize the effect of outlier measurements in the clustering procedure. The proportion of variance explained by time variables increased from < 24% before scaling to >80% after it. Several clustering methods including hierarchical clustering, c-means soft clustering and STEM where tested, all of which had similar results (every cluster had > 50% one-to-one correspondence across algorithms). The *k* parameter (*k* = 7) was chosen by the elbow method on the basis of functional (total number of enriched GO terms) and structural criteria (percentage of variance explained) readouts as shown in Supplementary Fig. 2d. To quantify per-cluster enrichment of gene annotation sets, we used the enrichment test based in the hyper-geometric distribution calculated by the R Bioconductor package clusterProfiler with Benjamini–Hochberg, FDR < 0.01[70]. GSEA[20] was used to analyze whether a previously defined gene sets show statistically significant, concordant differences between two biological states. ISMARA[16] was used for the prediction of retinal EC transcriptional regulators using the mm9 version of the mouse genome. The overall contribution of a TF motif was reported as a *z*-value that represents the average number of standard deviations of the motif activity from the zero across the time course. MamPhEA was used to evaluate the overrepresentation of mouse-mutant phenotypes[29].

**Cell culture and lentiviral infection.** MS1 mouse pancreas ECs (CRL-2279; ATCC) were cultured in DMEM medium supplemented with 5% fetal calf serum (FCS) at 37 °C and 5% CO$_2$. For extracellular stimulation, ECs were incubated for 2 h in serum-free EBM-2 (Lonza) followed by treatment of complete EGM-2 media (EBM-2+EGM-2 SingleQuot Kit; Lonza), hVEGF-A (20 ng/ml), hFGF-B (10 ng/ml), R$^3$-IGF-1 (10 ng/ml), or hEGF (10 ng/ml). HA or FLAG-tagged MafB construct was amplified by PCR, sub-cloned into pCR8/GW/TOPO (Invitrogen), and cloned into pLEX_307 (Addgene, #41392) by an LR recombination reaction. To produce lentiviruses, HEK293 cells were plated at a density of 8 × 10$^6$ per 10 cm dish. On the next day, the cells were transfected with 3 µg of psPAX2 (Addgene, #12260), 1.5 µg of pMD2.G (Addgene, #12259) and 4.5 µg of lentiviral vector pLKO.1-scrambled shRNA (Control), pLKO.1-Mafb-shRNA (TRCN0000081969), pLEX_307-mock, pLEX_307-HA-Mafb, pLEX_307-FLAG-Mafb, LeGO-E (eGFP, Addgene, #27359), or LeGO-T2 (tdTomato, Addgene, #27342) using 27 µl Fugene 6 in 600 µl Opti-MEM per dish. Virus-containing supernatants were collected at 24 h, filtered through a 0.4-µm PVDF filter (Millipore), concentrated at 20,000 rpm for 2 h using a optima L-100 XP ultracentrifuge (Beckman Coulter), re-suspended in knockout DMEM and then stored at −80 °C until use. The lentivirus-infected cells were then subcultured and selected in puromycin (Sigma) or blasticidin (Calbiochem) to generate stable cell lines. Control (scrambled) or MafB KD MS1 cells were transfected with Rac1-V12-Myc constructs for 24 h by PEI 25kD (Polysciences). The cell lines were not tested for mycoplasma contamination.

**3D fibrin gel bead sprouting assay in vitro.** The in vitro sprouting assay using 3D fibrin gel was performed as described previously[71] with minor modifications. In brief, MS1 mouse ECs were suspended with Cytodex 3 microcarriers (GE Healthcare Life Sciences) at a concentration of 400–450 cells per bead in 1 ml of DMEM growth medium (Sigma), incubated for 4 h at 37 °C with gentle shaking every 20 min, transferred to a low attachment surface dish, and incubated for overnight in 5 ml of DMEM at 37 °C and 5% CO$_2$. The following day, the cell-coated beads were resuspended in 3 mg/ml of fibrinogen (Sigma), received 0.625 U/ml of thrombin (Sigma) and were placed in 35 mm µ-Dish culture plates (Ibidi). The fibrinogen/bead solution was allowed to clot for 5 min at room temperature before it was placed at 37 °C and 5% CO$_2$ for 30 min. Growth media was added and equilibrated with the fibrin clot for 30 min and then replaced with 1 ml of fresh medium. C3H/10T1/2 fibroblasts (CCL-226; ATCC) were seeded on top of the fibrin gel at a concentration of 3 × 10$^4$ cells per dish. Growth medium was changed every other day. We monitored the number of EC sprouts for 3 days and counted 12 beads per condition. For the competitive sprouting assay, we mixed differentially labeled ECs with eGFP or tdTomato at a 1:1 ratio and then analyzed at 8 h after the fibroblast seeding.

**Quantitative RT-PCR.** Total RNA was isolated using RNeasy Micro plus kit (Qiagen) according to the manufacturer's protocol, and then used to generate cDNA with the iScript cDNA Synthesis kit (Bio-Rad). Real-time PCR was performed using TaqMan or SYBR green gene expression assays (Thermo) on an ABI PRISM 7900HT Sequence Detection System (Applied Biosystems) or CFX96 Real-time PCR Detection System (Bio-rad). Gene expression assays were normalized to endogenous *Actb* probe (4352341E; Thermo) as standard. FAM-conjugated Taq-Man probes for *Mafb* (Mm00627481_s1; Thermo), *Git1* (Mm01241200_m1; Thermo), and *Arhgdib* (Mm00801450_m1; Thermo) were used along with TaqMan Gene Expression Master Mix (Applied Biosystems) to perform qPCR. Relative quantification was carried out using ΔΔCT method.

**ChIP and ChIP-Seq analysis.** ChIP was performed as described previously[72] with some modifications. In brief, HA-tagged MafB overexpressed MS1 ECs were cross-linked with 1% (wt/vol) formaldehyde for 10 min at room temperature and then added glycine to a final concentration of 0.15 M. Chromatin was sonicated to an average fragment length of 150–250 bp using SFX250 digital sonifer (Branson) for 2 min: 0.7 s on/1.3 s off on ice and immunoprecipitated with anti-HA antibody-conjugated magnetic beads (M180-11; MBL) overnight at 4 °C. Following protein–DNA complex elution, samples were treated with RNaseA and proteinase K and incubated at 65 °C overnight for reverse cross-linking. Immunoprecipitated

DNA was purified using AMPure XP SPRI beads (Beckman Coulter). Sequencing libraries were prepared with NEBNext Ultra DNA Library Prep Kit for Illumina (New England BioLabs) according to the manufacturer's instructions and sequenced on a MiSeq sequencer using $2 \times 75$ bp paired-end MiSeq v3 chemistry (Illumina). Sequencing reads were aligned to the mm10 reference mouse genome using Bowtie2[73] and analyzed with HOMER[31] for peak finding, motif analysis and peaks annotation. Gene set annotation enrichment tests were performed using GREAT v.3.0[32] with default parameters and were subsequently used for calculating the overlap with DEGs in the Mafb-cKO RNA-Seq experiment.

**Rac1/Cdc42 activation assay.** Control (scrambled) or MafB-KD MS1 ECs were serum starved for 2 h and stimulated with complete EGM-2 media for 30 min. Active Rac1 and Cdc42 were analyzed with Rac1/Cdc42 Activation Assay Kit (Millipore) according to the manufacturer's instructions. An aliquot of 100 μM of GTPγS or 1 mM of GDP were added to cell extracts and incubated at 30 °C for 15 min with agitation for positive or negative control, respectively.

**In vitro assays.** Chemotactic directional migration of mouse ECs exposed to a VEGF gradient (max. 50 ng/ml) was measured using Collagen IV-coated μ-Slide Chemotaxis 2D chamber (Ibidi) according to the manufacturer's instructions. Real-time images were taken with a fluorescence microscope Axio Observer Z1 (Carl Zeiss) equipped with a ×10 objective lens and an environmental chamber that maintained 37 °C and 5% $CO_2$ for 24 h at 10 min intervals and analyzed using Chemotaxis and Migration Tool (Ibidi) on ImageJ.

Scratch wound healing assay was performed to examine collective migration of ECs. Confluent cell layers in 24-well plates were carefully wounded using a sterile 200 μl tip, washed twice with medium and cultured for 12 h or 24 h. The area between the wound edges was measured before and after the recovery and quantified using light microscopy and ImageJ. For cell adhesion assay, MS1 cells were deprived of serum for 8 h, seeded to 0.02% Collagen I coated wells, and incubated for 30 min. After washed four times with media, adherent cells were cultured for 4 h for recovery, added by CCK-8 solution (Dojindo laboratories), then incubated for an additional 2 h. Absorbance at 490 nm were measured using BioPhotometer (Eppendorf).

**Statistics and reproducibility.** Data sets with normal distributions were analyzed with unpaired Student's two-tailed t-tests to compare two conditions. Results are depicted as mean ± s.d. or mean ± s.e.m. as indicated in figure legends. All experiments for quantitative analysis and representative images were reproduced at least three times.

**Data availability.** Sequencing data that support the findings of this study have been deposited in GEO under accession code GSE86788. All other data supporting the findings are available within the paper and its supplementary information files.

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

## Acknowledgements

This work was supported by the Max Planck Society, the University of Münster and the DFG cluster of excellence "Cells in Motion". B.H.-R. received support from the European Union's Horizon 2020 research and innovation program under the Marie Sklodowska-Curie grant agreement No. 643062—ZENCODE-ITN.

## Author contributions

R.H.A. and J.M.V. supervised the project. H.-W.J. conceived the project and performed experiments of RiboTag, RNA-Seq, RT-qPCR, immunostaining, ChIP-Seq, in vitro assays, and some bioinformatics analyses. B.H.-R. performed most of bioinformatics analyses. J.K. and J.Y. performed in vitro assays and molecular biological experiments. K.-P.K. performed lentiviral experiments. R.E.-G. provided bioinformatics pipelines. S.A. generated Cdh5-EGFP transgenic reporter mouse. H.R.S. supported the project. H.-W.J., B.H.-R, J.M.V. and R.H.A. wrote the manuscript.

## Additional information

**Competing interests:** The authors declare no competing financial interests.

