## [Peer Review File · Nature Communications]

Reviewers' comments:

Reviewer #1 (Remarks to the Author):

The study by Jeong et al reports the use of an elegant expression based approach to identify transcriptional regulators of angiogenesis using the mouse retina model. They generated a vascular specific Ribo-tag mouse model that allows the analysis of vessel specific transcriptional changes from whole retina extractions. They convincingly show that this approach works and apply it to characterize transcriptional changes in the retinal vasculature during postnatal retinal development. Focusing on transcription factors, they further investigate the role of MafB in the retina, which includes in vitro, excellent in vivo (retina) and transcriptomic analysis (both MafB ChIP seq and RNAseq from their in vivo model). Ultimately, they suggest that MafB regulates angiogenic sprouting by controlling Rac1, Cdc42 activity by the transcriptional regulation of Git1 and Arhgdbid. The later links and mechanistic work on MafB are somewhat weaker than the very useful resource of transcriptomic data presented.

The study will be of interest to vascular biologists and the angiogenesis field. The datasets that are described, when made publically available, will represent a useful resource for the angiogenesis community and groups studying MafB. As such the study will be of interest to the readers of Nature Comms. However, as it stands the manuscript requires several major and minor revisions. See below.

Major

1. The finding of a role for MafB in the vasculature is not unexpected given the 2015 papers from Koltowska and Dieterich, however the suggestion of a broad role in angiogenesis downstream of VEGFA does not fit with these previous findings. In the Dieterich study, the MafB KO mice displayed a normal development and hence normal cardiovascular development with only very mild defects in dermal lymphatics at stages as late as E14.5. In zebrafish trunk vessels mafb(a) was not expressed in arteries.

Is the role of MafB shown here only relevant to the retina or in postnatal settings? Are other tissues also showing defective angiogenesis in development or postnatal stages?

How do the authors explain the normal embryonic development of previously reported KO mice given the severe defects such as seen here in deep layer vessels?

At some points the paper seems to suggest that MafB regulates EC migration and behaviour generally but this is very hard to marry with published KO data.

2. Antibodies that detect MafB in vivo are reported (Dieterich et al 2015) and showed clearly lymphatic restricted expression in the developing skin. How does MafB localise in the developing retinal vasculature? Is it tip-cell enriched as would be expected if it is VEGFA responsive in vivo? Is it only found in blood vessels in the retina or more broadly in blood vessels in other tissues?

3. The relationship between MafB and Rac1, cdc42 is suggested by reduced activities with western blots but is not functionally delineated with epistasis. Do either or both activated Rac1 or Cdc42 rescue the sprouting and migration defects in vitro in the absence of MafB?

4. At many points throughout this study the authors did not show any n-values for how many biological and technical replicates the data sets represent. This should be made clearer throughout.

5. Is MafB downregulated at the protein level on KD and up-regulated in OE settings as expected?

Minor

6. The authors observe decreased proliferation in the *Mafb* Δ EC retinas, however in the RNA-seq data set the up-regulated genes were associated with cell cycle, mitosis etc. Is there a functional explanation as to why these genes are up-regulated? Wouldn't the authors expect to observed increased proliferation based on transcriptomic data? Does *Mafb* act to inhibit or promote EC progress through cell cycle based on the transcriptomics?

7. To clarify the text. The authors focus their analysis on 3,248 genes from cluster 5 and 6 – are these genes only enriched at the P10-P15 or enriched in all stages?

8. To clarify the experimental design and flow. It would be helpful if the authors explain a bit more about selection of *mafb* – from 93 transcription factors *Mafb* was in the top 30% of the hits – was it the only transcription factor in these top 30%? A table comparing the top hit transcription factors and their expression properties and clustering would be helpful.

Reviewer #2 (Remarks to the Author):

Jeong et al combined tissue-specific RiboTag technology and RNA-seq analysis to determine the temporal profiles of gene expression in the developing retinal vasculature. Their results revealed developmental stage-dependent activation of biological pathways and interesting transition between developmental stages. They then carried out more detailed analysis on one of the identified transcription factors *Mafb*, and demonstrated that some aspects of the temporal gene regulation were achieved by coordinated transcriptional activation and repression. Organ development is a step-wise process, which involves complex temporal control of multiple biological pathways. Describing the molecular basis of the temporal control is an important aspect in understanding organ development, and it requires work like this one. Therefore, I believe this manuscript is of interest to the general readership and is suitable for publication in Nature Communication after the authors substantiate a few technical points and add more mechanistic insights. Below are recommendations to improve on these two areas.

Major points:

1. The expression profiling results critically depend on the assumption that HA-labeling per endothelial cell is consistent between different developmental stages. This is assumed but not directly demonstrated. A straightforward way to confirm this is to quantify HA and a few stably-expressed EC-specific proteins, then normalize HA to these EC-specific proteins. Western blot or ELISA could be used for this purpose. If "HA/EC marker ratios" differ between different stages, the gene expression data at a given stage should be normalized to this denominator at that stage. In addition, the selected stably expressed EC-specific proteins should be plotted against stages to assure they all consistently reflect the increase in cell numbers as development proceeds.

2. Documenting temporal control of EC gene expression is very important, but to understand what it reflects is a requisite in understanding the biological meaning of the data. The authors used *Mafb* as an example of interesting temporal control of EC gene expression, I recommend that they also include co-staining with *Mafb* and an pan EC marker. This will allow them to better describe their data and distinguish the following potential cellular causes: changes in the number of ECs that express *Mafb*; switch in different populations of ECs that express *Mafb*; all ECs express *Mafb* equally but the per cell levels fluctuate; some combination of the above.

3. The authors described the function of *Mafb* using in vitro sprouting assay, as this assay integrated

many different cellular behaviors, it shed little light on the specific function(s) of Mafb. They should analyze Mafb activity in EC proliferation in response to different growth stimuli, EC survival, EC - connectivity, and collective migration, in addition to the in vitro directional migration experiment and in vivo EdU labeling they already did and reported later in the manuscript. These data will help us better understand the drivers of the in vivo phenotype, and potentially shed light on some of the temporal changes that are counter intuitive. For example, sprouting activity is very strong at P6, but much less at P15. If Mafb is directly regulating sprouting, why expression at P6 is lower? Perhaps Mafb has a more fundamental function that contributes to sprouting angiogenesis, instead of directly controlling sprouting.

4. The authors described a strong deficit in the formation of deeper vascular plexus in Mafb Δ EC retinas (Fig. 4c, d). Since many cellular processes contribute to the formation of the deeper plexus, the authors should provide a more detailed description of the phenotype: do the vertical sprouts ever form? If so, do they grow all the way to the outer plexiform layer (OPL)? If so, do they make the proper turns but ultimately failed to ramify horizontally in this layer? Analysis of confocal images of the retinas they already collected should be sufficient to answer these questions. Their finding might help to better elucidate the role of Mafb in sprouting angiogenesis.

5. To strengthen the proposed molecular pathway depicted in Fig. 7g, the authors should carry out Git1 overexpression and/or Arhgdib knockdown in Mafb deficient cells to determine if the phenotype could be alleviated. As it stands now, the evidence for the proposal is weak.

Minor points:

1. On page 8, lines 155-156: I suggest changing the statement to "... All of which indicates the functional relevance of the structured transcriptional response" because expression profiling alone is never sufficient to confirm functional relevance.

2. Fig. 4c, d: since the inner plexiform layer has not formed yet at p10 and p11, it would be nicer to label the bottom panels as "OPL" instead of "Deep".

3. It would be nice to define a new term when it is introduced. For example, I don't understand what a "developmental checkpoint (page 2, line 29)" is in the context of retinal vascular development. Please define the term, and explain the specifics --- i.e. what stage(s) are considered checkpoints in the neonatal retinal vasculature, what mechanism works to ensure one stage does not progress into the next if a checkpoint exists.

Reviewer #3 (Remarks to the Author):

This manuscript by Jeong et al. describes the use of contemporary experimental and computational techniques to explore gene expression in endothelial cells during angiogenesis. The germinal results for this study were obtained with mouse retina in situ using the RiboTag methodology, generating a picture of the translated transcriptome specifically in endothelial cells during postnatal development. Informatics analysis of the results allowed the authors to hypothesize candidate regulators. Further experimental studies, both in vivo and in vitro, using RNA-seq and ChIP-seq identified the transcription factor, Mafb, as a putative regulator of angiogenesis. They go to describe the suite of genes that is regulated in this context. This paper is of interest, not only specifically to vascular biologists, but to the community generally as a model of integrative analysis of gene expression in a complex biological system.

I have only a few minor criticisms or suggestions:

1. When the TMX-inducible Cre was employed, I presume that the controls were also injected with TMX. Perhaps this was stated explicitly and I missed it, but if not it should be stated.
2. Immunoprecipitation of specifically tagged ribosomes showed a 100-200 fold enrichment of endothelial-specific transcripts. Is this consistent with the relative content of this cell type in postnatal retina. If so, it should be stated and referenced. If not, what is the explanation?
3. The authors should abstain from unnecessary hyperbole in describing their results; for example on page 10 they refer to *Mafb* as being "strongly upregulated", when in fact the level of induction is less than 2-fold at a single time point. This overstatement of a single result detracts from the overall conclusions of the integrated approach.

First of all, we would like to thank the reviewers for their time and helpful comments, which have allowed us to improve the manuscript. Please find below our detailed point-by-point response to all questions and comments:

Reviewer #1 (Remarks to the Author):

The study by Jeong et al reports the use of an elegant expression based approach to identify transcriptional regulators of angiogenesis using the mouse retina model. They generated a vascular specific Ribo-tag mouse model that allows the analysis of vessel specific transcriptional changes from whole retina extractions. They convincingly show that this approach works and apply it to characterize transcriptional changes in the retinal vasculature during postnatal retinal development. Focusing on transcription factors, they further investigate the role of MafB in the retina, which includes in vitro, excellent in vivo (retina) and transcriptomic analysis (both MafB ChIP seq and RNAseq from their in vivo model). Ultimately, they suggest that MafB regulates angiogenic sprouting by controlling Rac1, Cdc42 activity by the transcriptional regulation of Git1 and Arhgd1d. The later links and mechanistic work on MafB are somewhat weaker than the very useful resource of transcriptomic data presented.

The study will be of interest to vascular biologists and the angiogenesis field. The datasets that are described, when made publically available, will represent a useful resource for the angiogenesis community and groups studying MafB. As such the study will be of interest to the readers of Nature Comms. However, as it stands the manuscript requires several major and minor revisions. See below.

Reply: We are grateful for this assessment of our work. As suggested by the reviewer, we have improved our manuscript and data regarding the mechanistic role of MafB.

Major

Question 1: The finding of a role for MafB in the vasculature is not unexpected given the 2015 papers from Koltowska and Dieterich, however the suggestion of a broad role in angiogenesis downstream of VEGFA does not fit with these previous findings. In the Dieterich study, the MafB KO mice displayed a normal development and hence normal cardiovascular development with only very mild defects in dermal lymphatics at stages as late as E14.5. In zebrafish trunk vessels mafb(a) was not expressed in arteries. Is the role of MafB shown here only relevant to the retina or in postnatal settings? Are other tissues also showing defective angiogenesis in development or postnatal stages? How do the authors explain the normal embryonic development of previously reported KO mice given the severe defects such as seen here in deep layer vessels? At some points the paper seems to suggest that MafB regulates EC migration and behaviour generally but this is very hard to marry with published KO data.

Reply: As the reviewer correctly states, the function of MafB in the blood vessel endothelium has not been investigated yet. *Mafb*^{-/-} mutants develop to term and die perinatally, which might indeed argue against a prominent role of the transcription factor in embryonic angiogenesis. Possible explanations for the lack of embryonic vascular defects include 1) functional redundancy for MafB with other large Maf or bZIP family proteins, and/or 2) tissue-specific regulation of MafB expression and function in endothelium. The large Maf proteins, MafA, MafB, c-Maf, and Nrl, have bZIP domains are highly homologous to a similar domain in other bZIP transcription factors. They recognize DNA motifs called Maf recognition elements (MAREs), which are derivatives of AP-1 and ATF/CREB consensus motifs. Although MafB is expressed widely in many different tissues including spleen, lymph node, bone marrow, heart, lung, liver, and skeletal muscle (Wang et al., *Genomics* 59:275, 1999), knockout phenotypes have been reported only in some of these tissues, suggesting that other large Maf or bZIP family proteins compensate for MafB in the regulation of target gene expression in a tissue- or cell type-specific manner. In addition, small Maf protein homodimers can compete with MafB for binding to target gene promoters (Eychene et al., *Nat Rev Cancer* 8:683, 2008). Thus, one would presumably have to study double or triple knockout mice to fully uncover the role of large Maf family proteins in growing blood vessels. This question would go beyond the scope of the present manuscript, in which we have investigated MafB as an example for the validation of RNA-seq data.

Regarding the expression of MafB in blood vessel endothelium, Koltowska and colleagues indicated that *mafba* was highly enriched in venous vs. arterial ECs at 60 hpf. The gene was also expressed in venous ECs by 5 dpf but was more enriched in lymphatic ECs (Koltowska et al., *Genes & Dev.* 2015, Fig. 3B & C). Likewise, Dieterich et al. (*Dev Cell* 2015, Fig. 3B) reported MafB expression in cultured human blood vessel ECs even though expression was comparably low relative to lymphatic ECs. Using RNA-Seq analysis (unpublished), we found that MafB expression in endothelium is highly tissue-specific. MafB is, for example, most highly expressed in bone ECs (Figure R1 for reviewers only) and preliminary data indicate the presence of vascular defects in this tissue (data not shown).

Furthermore, newly added data (Fig. S5a-d) supports that MafB protein is indeed expressed in the retinal vasculature. Immunostaining shows that sprouting ECs contain more nuclear MafB than the cells in the adjacent vessel plexus (Fig. S5a). In the more mature central vessel plexus, nuclear MafB decorates arterial ECs, whereas comparably weak signals can be seen in veins (Fig. S5d).

Finally, we would also like to point out that defects in the postnatal mutant retinal vasculature provide very strong evidence that MafB is indeed regulating blood vessel angiogenesis. Processes in the retina are also unlikely to be disturbed by potential lymphatic defects in other organs and tissues.

Question 2. Antibodies that detect MafB *in vivo* are reported (Dieterich et al 2015) and showed clearly lymphatic restricted expression in the developing skin. How does MafB localise in the developing retinal vasculature? Is it tip-cell enriched as would be expected if it is VEGFA responsive *in vivo*? Is it only found in blood vessels in the retina or more broadly in blood vessels in other tissues?

Reply: Following the reviewer's suggestion, we performed immunostaining against MafB in developing retina of P6 Cdh5-EGFP transgenic reporter mice, a line that expresses nuclear EGFP is specifically in endothelial cells. This approach revealed that sprouting ECs contain more nuclear MafB than the cells in the adjacent vessel plexus (Fig. S5a), which is consistent with the induction of MafB by VEGF. In the more mature central vessel plexus, nuclear MafB can be predominantly seen in arterial ECs (Fig. S5d). In addition to the reported induction of MafB expression by VEGF-A and VEGF-C in lymphatic endothelial cells (Dieterich et al., Cell Reports 13:1493, 2015, Koltowska et al., Genes & Dev. 29:1618, 2015), experiments in K562 erythroleukemia cells have placed ERK signaling upstream of MafB (Sevinsky et al., MCB 24:4534, 2004). ERK signaling is also known to play an important role in arterial morphogenesis (e.g. Deng et al., Blood 121: 3988–3996), which is consistent with the arterial MafB signal seen *in vivo* (Fig. S5d).

In addition, MafB is expressed in other (non-vascular) cells, which we have, however, not investigated further because this aspect lies outside of the scope of the present study.

Question 3. The relationship between Mafb and Rac1, cdc42 is suggested by reduced activities with western blots but is not functionally delineated with epistasis. Do either or both activated Rac1 or Cdc42 rescue the sprouting and migration defects *in vitro* in the absence of MafB?

Reply: As requested by the reviewer, we have investigated this aspect in greater detail. Expression of constitutively active Rac1 (Rac1-V12) was able to rescue migration and sprouting defects in Mafb-KD MS1 endothelial cells (Fig. S8a-d).

Question 4. At many points throughout this study the authors did not show any n-values for how many biological and technical replicates the data sets represent. This should be made clearer throughout.

Reply: We now provide this information in the figure legends, as requested by the reviewer.

Question 5. Is MafB downregulated at the protein level on KD and up-regulated in OE settings as expected?

Reply: We now provide Western blot showing the expected up- and down-regulation of MafB protein in the OE and knockdown experiments, respectively (Fig. S4a).

Minor

Question 6. The authors observe decreased proliferation in the Mafb^{-/-} EC retinas, however in the RNA-seq data set the up-regulated genes were associated with cell cycle, mitosis etc. Is there a functional explanation as to why these genes are up-regulated? Wouldn't the authors expect to observed increased proliferation based on transcriptomic data? Does Mafb act to inhibit or promote EC progress through cell cycle based on the transcriptomics?

Reply: The enrichment of cell cycle-associated genes in Mafb cKO ECs does not automatically reflect an increase in proliferation. In fact, this gene set includes also many inhibitors of the cell cycle. Loss of MafB leads, for example, to upregulation of the tumor suppressors Rb1 and Pten, which can negatively regulate EC proliferation.

Question 7. To clarify the text. The authors focus their analysis on 3,248 genes from cluster 5 and 6; are these genes only enriched at the P10-P15 or enriched in all stages?

Reply: Those 3,248 genes from cluster 5 and 6 were enriched at P10-P15, as shown in Fig 2b.

Question 8. To clarify the experimental design and flow. It would be helpful if the authors explain a bit more about selection of mafb; from 93 transcription factors Mafb was in the top 30% of the hits; was it the only transcription factor in these top 30%? A table comparing the top hit transcription factors and their expression properties and clustering would be helpful.

Reply : To identify novel candidate angiogenic factors from the retinal EC transcriptome data set, we used two independent computational approaches, namely DEG clustering and ISMARA analyses. The former identified 3,248 genes in the angiogenic clusters 5 and 6, whereas ISMARA identified 93 transcription factors showing significant motif activities across the different stages. By integrating these two data sets, we selected 19 genes as the strongest candidates (see Fig. 2f). This list includes known angiogenic TFs such as Hif1a, Sox17, Lmo2, and Elf1/2/4 but also several novel potential regulators of blood vessel growth (such as MafB).

Reviewer #2 (Remarks to the Author):

Jeong et al combined tissue-specific RiboTag technology and RNA-seq analysis to determine the temporal profiles of gene expression in the developing retinal vasculature. Their results revealed developmental stage-dependent activation of biological pathways and interesting transition between developmental stages. They then carried out more detailed analysis on one of the identified transcription factors Mafb, and demonstrated that some aspects of the temporal gene regulation were achieved by coordinated transcriptional activation and repression. Organ development is a step-wise process, which involves complex temporal control of multiple biological pathways. Describing the molecular basis of the temporal control is an important aspect in understanding organ development, and it requires work like this one. Therefore, I believe this manuscript is of interest to the general readership and is suitable for publication in Nature Communication after the authors substantiate a few technical points and add more mechanistic insights.

Below are recommendations to improve on these two areas.

Reply: We are grateful for this assessment of our work. As suggested by the reviewer, we have improved our manuscript and data regarding the mechanistic role of MafB.

Major points:

Question 1. The expression profiling results critically depend on the assumption that HA-labeling per endothelial cell is consistent between different developmental stages. This is assumed but not directly demonstrated. A straightforward way to confirm this is to quantify HA and a few stably-expressed EC-specific proteins, then normalize HA to these EC-specific proteins. Western blot or ELISA could be used for this purpose. If HA/EC marker ratios differ between different stages, the gene expression data at a given stage should be normalized to this denominator at that stage. In addition, the selected stably expressed EC-specific proteins should be plotted against stages to assure they all consistently reflect the increase in cell numbers as development proceeds.

Reply: As the reviewer indicated, tagging (or labeling) efficiency of the ribosomal protein is critical for the immunoprecipitation and the following gene expression analysis. Especially in the case of the TRAP technique, a similar approach of ribosomal protein tagging using transgenic mice, the level of tagged ribosome depends on expression levels from cell-type-specific promoters that are quite variable in transcriptional activity (Doyle et al., Cell 135:749, 2008; Heiman et al., Cell 135:738, 2008). In RiboTag mice, however, expression of the HA-tagged ribosomal protein is not controlled by a transgenic allele but by a Cre-controlled modification of the endogenous Rpl22 locus. Thus, expression of Rpl22-HA protein in a specific cell type is neither variable nor reversible, which is one of the

most important advantages of the approach (also described in the original paper, second paragraph in the Discussion of Sanz et al., PNAS 106:13939, 2009). In our study, we induced the Cre-mediated recombination by injecting tamoxifen at P1 to P3 for the analysis of all different developmental stages. As a result, Rpl22-HA protein was robustly expressed in retinal endothelium as early as at P6 (Fig.1b) up to P50, the last stage analyzed (see new data added to Fig. S1a). Moreover, correlation coefficients generated by using the Spearman method for each RNA-Seq data further demonstrated the high consistency of RiboTag-RNA-Seq analyses between biological replicates or different developmental stages (see Figure R2).

The correlation between HA-Rpl22 and EC-specific marker proteins is, however, more complicated than it might appear at first glance. Transcript and gene product levels of EC markers, such as Pecam1, Endomucin and VE-cadherin, are not consistent between endothelial subpopulations or angiogenic conditions (Berger et al., J Cutan Pathol 20:399, 1993; Kusumbe et al., Nature 507:323, 2014; Monvoisin et al., Dev Dyn 235:3143, 2006). The lifetime of proteins, which depends on factors such as stability, degradation and recycling, is also highly distinct so that one cannot expect simple linear relationships between markers and EC number across different stages and conditions.

Question 2. Documenting temporal control of EC gene expression is very important, but to understand what it reflects is a requisite in understanding the biological meaning of the data. The authors used *Mafb* as an example of interesting temporal control of EC gene expression, I recommend that they also include co-staining with *Mafb* and an pan EC marker. This will allow them to better describe their data and distinguish the following potential cellular causes: changes in the number of ECs that express *Mafb*; switch in different populations of ECs that express *Mafb*; all ECs express *Mafb* equally but the per cell levels fluctuate; some combination of the above.

Reply: Following the reviewer's suggestion, we performed immunostaining against *MafB* in the P6 mouse retina (Fig. S5a-d). In this experiment, we used *Cdh5*-EGFP transgenic reporter mice in which nuclear EGFP is specifically expressed in ECs. As mentioned above in reply to a similar question raised by reviewer #1, sprouting ECs contain more nuclear *MafB* than the cells in the adjacent vessel plexus (Fig. S5a), which is consistent with the induction of *MafB* by VEGF. In the more mature central vessel plexus, nuclear *MafB* can be predominantly seen in arterial ECs (Fig. S5d). In addition to the reported induction of *MafB* expression by VEGF-A and VEGF-C in lymphatic endothelial cells (Dieterich et al., Cell Reports 13:1493, 2015, Koltowska et al., Genes & Dev. 29:1618, 2015), experiments in K562 erythroleukemia cells have placed ERK signaling upstream of *MafB* (Sevinsky et al., MCB 24:4534, 2004). ERK signaling is also known to play an important role in arterial morphogenesis (e.g. Deng et al., Blood 121: 3988–3996), which is again consistent with the arterial *MafB* signal seen *in vivo* (Fig. S5d).

Question 3. The authors described the function of *Mafb* using in vitro sprouting assay, as this assay integrated many different cellular behaviors, it shed little light on the specific function(s) of *Mafb*. They should analyze *Mafb* activity in EC proliferation in response to different growth stimuli, EC survival, EC-connectivity, and collective migration, in addition to the in vitro directional migration experiment and in vivo EdU labeling they already did and reported later in the manuscript. These data will help us better understand the drivers of the in vivo phenotype, and potentially shed light on some of the temporal changes that are counter intuitive. For example, sprouting activity is very strong at P6, but much less at P15. If *Mafb* is directly regulating sprouting, why expression at P6 is lower? Perhaps *Mafb* has a more fundamental function that contributes to sprouting angiogenesis, instead of directly controlling sprouting.

Reply: As suggested by the reviewer, we performed additional experiments to study the function of *MafB*. This revealed that collective migration (scratch wound assay) and cell adhesion to collagen I were significantly reduced in *Mafb* knockdown cells (Fig. S4c and d). In contrast, we observed no significant differences of proliferation and apoptosis in *Mafb* KD cells *in vitro* (data not shown). However, this most likely reflects that MS1 cells are immortalized by the SV40 large T antigen.

In vivo, sprouting of the wild-type retinal vasculature is increased after P6 because of the persistent angiogenesis in the outer and inner plexiform layers, and then dramatically decreased at P21. Indeed, the genes enriched in tip-cells such as *Dll4*, *Cxcr4*, *Angpt2*, and *Lcp2* were all included in the angiogenic cluster, cluster 5 and showed the highest expression at P10 and P15, while other tip-cell enriched genes, *Esm1* and *Apln*, were included in cluster 4 showing peak expressions at P10 (Table S2). Likewise, *Mafb* also showed the peak expression at P10 then decreased at P21 (Fig 3a), consistent with the tip cell-enriched expression of *MafB* and its role in regulating sprouting morphogenesis. New added supporting the role of *MafB* in endothelial cell migration has been added to Figure S4.

Question 4. The authors described a strong deficit in the formation of deeper vascular plexus in *Mafb* EC retinas (Fig. 4c, d). Since many cellular processes contribute to the formation of the deeper plexus, the authors should provide a more detailed description of the phenotype: do the vertical sprouts ever form? If so, do they grow all the way to the outer plexiform layer (OPL)? If so, do they make the proper turns but ultimately failed to ramify horizontally in this layer? Analysis of confocal images of the retinas they already collected should be sufficient to answer these questions. Their finding might help to better elucidate the role of *Mafb* in sprouting angiogenesis.

Reply: Our present findings suggested that the loss of Mafb in ECs leads to the sprout formation defect in mouse retina, as shown in Fig 4g. Vertical sprouts at P10 were also not formed properly in those mutant mice; as a result, deeper plexus in the OPL was dramatically reduced. We added the y-z plane depiction of the confocal images showing vertical sprouts at P10 in Figure S6c.

Question 5. To strengthen the proposed molecular pathway depicted in Fig. 7g, the authors should carry out Git1 overexpression and/or Arhgdib knockdown in Mafb deficient cells to determine if the phenotype could be alleviated. As it stands now, the evidence for the proposal is weak.

Reply: Previous studies already shown that Git1 and Arhgdib modulate cell migration via Rac1/Cdc42 activity regulation in opposite ways (Majumder et al., Arterioscler Thromb Vasc Biol. 34:419, 2014; Agarwal et al., Oncogene 32:2521, 2013). As the reviewer suggested, we tried to perform Git1 overexpression and Arhgdib knockdown in ECs using lentiviral infection combined with different selection markers. However, the knockdown of Arhgdib lead to cell death so that further functional analyses could not be performed. Instead, we transfected constitutively active form of Rac1 (Rac1-V12) into control (scrambled) or Mafb-KD MS1 endothelial cells to reactivate the small GTPase. As a result, the collective migration and sprouting defects in Mafb KD cells were significantly rescued to levels seen in Rac1-V12-expressing control cells (Figure S8).

Minor points:

Question 6. On page 8, lines 155-156: I suggest changing the statement to 'All of which indicates the functional relevance of the structured transcriptional response' because expression profiling alone is never sufficient to confirm functional relevance.

Reply: We have changed the text as suggested by the reviewer (confirms > suggests).

Question 7. Fig. 4c, d: since the inner plexiform layer has not formed yet at p10 and p11, it would be nicer to label the bottom panels as 'OPL' instead of 'Deep'.

Reply: Agree. We have modified the labels accordingly (Deep > OPL).

Question 8. It would be nice to define a new term when it is introduced. For example, I don't understand what a 'developmental checkpoint (page 2, line 29)' is in the context of retinal vascular development.

Please define the term, and explain the specifics --- i.e. what stage(s) are considered checkpoints in the neonatal retinal vasculature, what mechanism works to ensure one stage does not progress into the next if a checkpoint exists.

Reply: Agree. As suggested, we have replaced the term 'checkpoints' by 'stages' on page 2 and page 20.

Reviewer #3 (Remarks to the Author):

This manuscript by Jeong et al. describes the use of contemporary experimental and computational techniques to explore gene expression in endothelial cells during angiogenesis. The germinal results for this study were obtained with mouse retina in situ using the RiboTag methodology, generating a picture of the translated transcriptome specifically in endothelial cells during postnatal development. Informatics analysis of the results allowed the authors to hypothesize candidate regulators. Further experimental studies, both in vivo and in vitro, using RNA-seq and ChIP-seq identified the transcription factor, Mafk, as a putative regulator of angiogenesis. They go to describe the suite of genes that is regulated in this context. This paper is of interest, not only specifically to vascular biologists, but to the community generally as a model of integrative analysis of gene expression in a complex biological system.

Reply: We are grateful for this assessment of our work.

I have only a few minor criticisms or suggestions:

Question 1. When the TMX-inducible Cre was employed, I presume that the controls were also injected with TMX. Perhaps this was stated explicitly and I missed it, but if not it should be stated.

Reply: As suggested, we added the description "Tmx-injected Mafkbp/p littermates were always used as controls" to the Methods (section 'Mice and Inducible Genetic Modifications').

Question 2. Immunoprecipitation of specifically tagged ribosomes showed a 100-200 fold enrichment of endothelial-specific transcripts. Is this consistent with the relative content of this cell type in postnatal retina. If so, it should be stated and referenced. If not, what is the explanation?

Reply: As the reviewer pointed out, the fold-enrichment of a transcript after IP is in inverse proportion to the target cell abundance in whole tissue, but not fully consistent with the cellular composition because of the IP efficiency. According to a recent paper analyzed whole mouse retina using single cell RNA-Seq, the

percentage of endothelial cells in adult mouse retina is as low as 0.6% (Macosko et al., Cell 161:1202, 2015). At P6, the number of endothelial cells would be much less than that of in adult, presumably around 0.1 ~ 0.2%. Thus, 100 to 200 fold enrichment of endothelial cell marker genes after the IP indicated that the efficiency of IP against HA was approximately 10 ~ 40% (an approximate IP efficiency higher than 20% is generally acknowledged to be very high, while most are in the 1 ~ 20% range).

Question 3. The authors should abstain from unnecessary hyperbole in describing their results; for example on page 10 they refer to *Mafb* as being ‘strongly upregulated’, when in fact the level of induction is less than 2-fold at a single time point. This overstatement of a single result detracts from the overall conclusions of the integrated approach.

Reply: Agree. We have modified the text accordingly: On page 8 (line 9), ‘confirms’ was changed to ‘suggests’; and on page 10 (line 8), ‘strongly’ was changed to ‘significantly’.

REVIEWERS' COMMENTS:

Reviewer #1 (Remarks to the Author):

The authors have addressed all of my major concerns. The revised manuscript is much improved. The study represents a very useful resource for the angiogenesis field as well as offering considerable new insights into the role of MafB in vasculature.

The paper is thorough, well written and clear. I expect it will be well received by the vascular biology community.

Reviewer #2 (Remarks to the Author):

The authors answered most of my questions. Although some of my questions cannot be sufficiently addressed, the added data and discussion improved the manuscript sufficiently to qualify it for publication. I recommend acceptance of this manuscript for publication in Nature Communications.

Reviewer #3 (Remarks to the Author):

The authors responded appropriately to my critique and this reviewer (#3) requires no further revision.